# CREPT is required for murine stem cell maintenance during intestinal regeneration

Liu Yang [1], Haiyan Yang[2], Yunxiang Chu[3], Yunhao Song[1], Lidan Ding[1], Bingtao Zhu[1], Wanli Zhai[1],
Xuning Wang[4], Yanshen Kuang[4], Fangli Ren[1], Baoqing Jia[4], Wei Wu[2], Xiongjun Ye [5✉], Yinyin Wang [1✉] &
Zhijie Chang [1✉]

Intestinal stem cells (ISCs) residing in the crypts are critical for the continual self-renewal and rapid recovery of the intestinal epithelium. The regulatory mechanism of ISCs is not fully understood. Here we report that CREPT, a recently identified tumor-promoting protein, is required for the maintenance of murine ISCs. CREPT is preferably expressed in the crypts but not in the villi. Deletion of CREPT in the intestinal epithelium of mice (Vil-CREPT[KO]) results in lower body weight and slow migration of epithelial cells in the intestine. Vil-CREPT[KO] intestine fails to regenerate after X-ray irradiation and dextran sulfate sodium (DSS) treatment. Accordingly, the deletion of CREPT decreases the expression of genes related to the proliferation and differentiation of ISCs and reduces Lgr5[+] cell numbers at homeostasis. We identify that CREPT deficiency downregulates Wnt signaling by impairing β-catenin accumulation in the nucleus of the crypt cells during regeneration. Our study provides a previously undefined regulator of ISCs.

[1] State Key Laboratory of Membrane Biology, School of Medicine, Center for Synthetic and Systems Biology, Tsinghua University, 100084 Beijing, China. [2] MOE Key Laboratory of Protein Sciences, School of Life Sciences, Tsinghua University, 100084 Beijing, China. [3] Department of Gastroenterology, Emergency General Hospital, 100028 Beijing, China. [4] Department of Gastroenterology, Chinese PLA General Hospital, 100700 Beijing, China. [5] Urology and Lithotripsy Center, Peking University People's Hospital, 100034 Beijing, China. ✉email: urologye@sina.com; wangyinyin@tsinghua.edu.cn; zhijiec@tsinghua.edu.cn

The intestinal epithelium self-renews rapidly and turns over every 3–5 days[1–3]. The ISCs residing in intestinal crypts play an important role in intestinal renewal. ISCs are able to differentiate into all types of intestinal cells. One part of ISC daughter cells constitute a transit-amplifying (TA) cell compartment. TA cells migrate upwards along the crypt-villi axis and mature into absorptive and secretory cells in the villi[4]. These mature cells ultimately undergo apoptosis at the villus tip after 3–5 days. Another part of ISC daughter cells migrates downwards and differentiates into the Paneth cells at the bottom of crypts[5,6]. The Paneth cells finally detach and separate the ISCs individually[7]. The above processes of ISC proliferation and differentiation occur continually to maintain intestinal homeostasis.

Beyond daily renewal, the intestinal epithelium displays regeneration ability after damages induced by ionizing radiation and chemical reagents[8]. The ISCs are believed to facilitate intestinal regeneration. At least two types of ISCs exist in the intestine. Potten et al reported that slow-cycling and DNA-label-retaining cells, also known as position 4 or +4 cells, reside above Paneth cells[9,10]. The +4 cells, marked by highly expressed Bmi1, Hopx, mTERT, and Lrig1[11–14], are known as quiescent and "reserve" stem cells. These cells are radiation resistant and involved in injury responses, but not required for daily intestine renewal[15]. However, the role of +4 stem cells needs to be validated because Bmi1, Hopx, and Lrig1 are also mostly enriched in cells at the base of crypts, not only at the '+4' position[16]. Barker et al reported that Lgr5 (leucine-rich-repeat-containing G-protein-coupled receptor 5, also known as Gpr49), one of Wnt signaling receptors, was specifically expressed in the crypt base columnar cell (CBC) and could be a marker for stem cells since Lgr5+ cells were able to generate all epithelial lineages. Lgr5+ cells have been widely considered as a representative of stem cells of intestines[2]. Metcalfe et al proved that Lgr5+ stem cells but not +4 cells were irradiation resistant and indispensable for intestinal regeneration after radiation damage[17]. Although these different ISCs populations have been identified, however, the underlying mechanisms as to how these ISCs maintain homeostasis and promote regeneration remain elusive.

Accumulating evidence suggests that the Wnt signaling pathway is essential for the ISC self-renewal and proliferation of crypt compartment[1,18,19]. When Wnt ligands encounter their Frizzled-Lrp5/6 receptors, β-catenin is stabilized from APC destruction complex, accumulates in the cytoplasm, and travels to the nucleus. Nuclear β-catenin activates Wnt target genes by interacting with transcription factor TCF[20,21]. Currently, a wide set of Wnt targeted genes have been identified in ISCs. Importantly, regulators for Wnt signaling have also been demonstrated to play an important role in the maintenance of ISCs[22].

CREPT, also named RPRD1B, was identified as a tumor-related gene because of its upregulation in tumors[23]. Accumulating studies demonstrated the role of CREPT in tumors[24–26]. Deletion of CREPT impeded tumorigenesis but overexpression of CREPT promoted tumor formation aggressively[23]. CREPT was demonstrated to upregulate cyclin D1 and cyclin B1 expression by promoting cell cycle at both G1 and G2 phases[27]. Our previous studies showed that CREPT participated in Wnt/β-catenin signaling by binding to the β-catenin/TCF4 complex and promoting the transcription of Wnt target genes in tumor cells[28,29]. Other groups also demonstrated that CREPT associated with RNA polymerase II and S5-phosphatase RPAP2 to regulate gene transcription[30–32]. However, previous studies of CREPT mainly focused on signaling regulation in tumor cells. Little is known about the role of CREPT in normal tissues.

In this study, we report that CREPT is mainly expressed in intestinal crypts, where the ISCs reside. CREPT deletion decelerates the fast turnover of intestinal epithelia. CREPT deficient intestinal epithelia fail to recover from X-ray radiation and dextran sulfate sodium (DSS) treatment. Furthermore, CREPT deletion leads to a substantial drop in the number of Lgr5+ ISCs and substantial downregulation of proliferation and differentiation genes in the ISCs at homeostasis. In addition, CREPT is required for Wnt activation by facilitating nuclear β-catenin retention in the ISCs. Our data identify CREPT as a regulator for Lgr5+ ISCs to maintain homeostasis and a Wnt signaling activator during intestinal regeneration.

## Results

**CREPT deficiency decelerates the daily renewal of intestinal epithelium.** To study the role of CREPT in intestine maintenance, we deleted CREPT specifically in the intestinal epithelium (assigned Vil-CREPT$^{KO}$) by introducing *villin-cre*[33] into *CREPT$^{fl/fl}$* mice (Supplementary Fig. 1a). A Western blot demonstrated that CREPT was effectively deleted in the small and large intestines (Supplementary Fig. 1b). The low level of CREPT protein as detected by the Western blot might be due to the expression in non-epithelial cells in the intestine. A histological analysis showed that CREPT was completely deleted in the epithelial cells of small (Fig. 1a) and large intestines (Supplementary Fig. 1c) but remained observable levels in the connective tissue cells of Vil-CREPT$^{KO}$ mice. Of note, CREPT protein is mainly expressed in the nuclei of crypt cells of wild type (WT) mice (Fig. 1a). A qRT-PCR analysis confirmed that the expression of CREPT is abundant in the crypts, similar to that of Olfm4, a marker of crypts (Fig. 1b).

Vil-CREPT$^{KO}$ mice were not born at the expected Mendelian ratio (Supplementary Fig. 1d). The survived Vil-CREPT$^{KO}$ adult mice showed substantially decreased body size (Fig. 1c) and significant body weight loss (Fig. 1d). IHC analyses showed that the villus number and structure in the small intestine were normal (Supplementary Fig. 1e, top panel), but, the crypts in the large intestine were enlarged and became thick (Supplementary Fig. 1e, bottom panel) in the Vil-CREPT$^{KO}$ mice. To address whether CREPT affects the status of epithelial cells, we stained the cells with Ki67, a widely used marker for cell proliferation. The results showed that Ki67 positive cells distributed in the villi were substantially reduced in Vil-CREPT$^{KO}$ mice while those cells in the crypts appeared no difference between WT and Vil-CREPT$^{KO}$ mice (Fig. 1e). Interestingly, the number of Ki67 positive cells was decreased in the crypts in the large intestine of Vil-CREPT$^{KO}$ mice in comparison with WT mice (Supplementary Fig. 1f). To address whether the deletion of CREPT influences cell proliferation, we pulse-labeled the proliferating cells with EdU (5-ethynyl-2′-deoxyuridine)[34]. The results showed that EdU-labeled cells were restricted to the crypt compartment in intestines after 2 h incorporation in both WT and Vil-CREPT$^{KO}$ mice (Fig. 1f, top panel). However, the number of EdU-labeled cells was significantly decreased in Vil-CREPT$^{KO}$ mice (Fig. 1g), which corresponded with a substantial decrease of S/G2/M phase cell numbers in Vil-CREPT$^{KO}$ mice (Supplementary Fig. 1g). Furthermore, we observed that EdU+ cells migrated slowly in Vil-CREPT$^{KO}$ mice in comparison with WT mice at 24 h after EdU labeling (Fig. 1f, middle panel, and 1 h). The EdU+ cells migrated to the top of villi in WT mice but were not extruded on the crypts-villi axis in Vil-CREPT$^{KO}$ mice at 72 h after labeling (Fig. 1f, bottom panel, and 1i). The EdU+ cells showed similar patterns in the distribution at the large intestines in both WT and Vil-CREPT$^{KO}$ mice at 24 and 48 h after EdU labeling (Supplementary Fig. 1h). As for differentiated intestinal cells, the number of lysozyme stained Paneth cells (Fig. 1j) of small intestines and alcian blue stained goblet cells of small (Fig. 1k, l) and large (Supplementary Fig. 1i) intestines were significantly

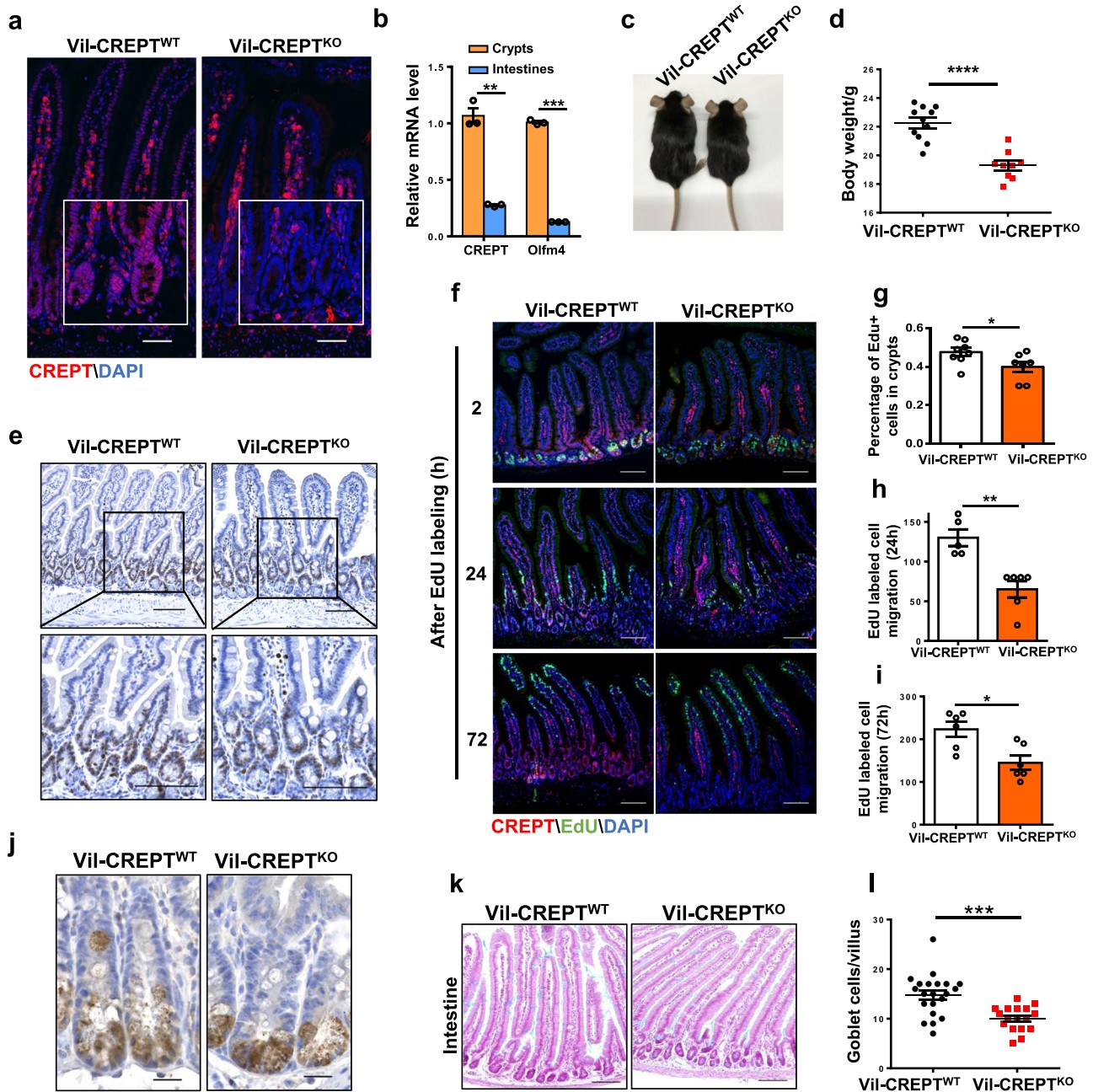

**Fig. 1 CREPT is required for maintaining the rapid turnover of the intestinal epithelium. a** Representative images of fluorescent CREPT staining in small intestines from WT and Vil-CREPT[KO] mice. Images are representative of n = 4 mice per genotype. **b** Quantitative RT-PCR analysis of CREPT and Olfm4 in crypt cells and epithelial cells (crypt + villous cells). p = 0.0067 (CREPT); p = 0.0002 (Olfm4). n = 3 independent experiments. **c** Representative images of WT and Vil-CREPT[KO] mice. **d** Quantification of body weights of WT and Vil-CREPT[KO] mice. n = 11 WT mice and n = 9 Vil-CREPT[KO] mice. p < 0.0001. **e** Representative images of Ki67 staining in duodenums of WT and Vil-CREPT[KO] mice. Images are representative of n = 4 mice per genotype. **f** Representative confocal images of cell migration in WT and Vil-CREPT[KO] mouse at 2-h, 24-h, and 72-h after EdU injection. Images are representative of 3 independent experiments (n = 6 mice per genotype). **g** Quantification of EdU+ cell number in each crypt 2 h after EdU labeling. n = 3 independent experiments. p = 0.0435. **h** Quantification of EdU+ cells migration length at 24 h. The migration distance of the fastest cells was measured from the bottom of the crypts. n = 3 independent experiments. p = 0.0017. **i** Quantification of Edu+ cell migration length at 72 h. The migration distance of the slowest cells was measured from the bottom of the crypts. n = 3 independent experiments. p = 0.0412. **j** Representative images of Paneth cells staining in duodenums of WT and Vil-CREPT[KO] mice. n = 4 mice per genotype. **k** Representative images of goblet cells staining in the jejunum of WT and Vil-CREPT[KO] mice. n = 3 mice per genotype. **l** Quantification of goblet cell number per villus. n = 3 mice per genotype. p = 0.0002. Statistics data represent mean ± SEM. All p values were generated by 2-tailed Student's t-test. Scale bars: 20 μm (**a** and **j**), 100 μm (**e**, **f**, **k**).

decreased in Vil-CREPT[KO] mice. Taken together, these results suggest that the deletion of CREPT in epithelial cells leads to an abnormal crypt structure in the large intestine, slower migration of epithelial cells in villus in the small intestine, and lower number of proliferating cells in the small intestine.

**CREPT deficiency in the epithelium decreases Lgr5+ ISCs and impairs organoid formation.** Since CREPT deletion decelerated the turn-over rate of intestinal epithelial cells and decreased the number of proliferated and differentiated cells, we questioned if CREPT deletion might impair the function of ISCs. To determine

whether a specific type of stem cell responds to CREPT deletion, we examined the putative ISC markers. The CBC stem cell markers (Lgr5 and Olfm4)[2,35] were downregulated in the crypts of Vil-CREPT[KO] mice, whereas reserved stem cell markers (Dclk1 and Bmi1)[13,36], and both CBC and reversed ISC markers (Sox9 and Lrig1)[12,37] were barely affected (Fig. 2a). Indeed, the number of Lgr5-GFP[+] stem cells was significantly decreased in *Villin-cre; Lgr5-EGFP; CREPT[fl/fl]* mice (Fig. 2b, c, Supplementary Fig. 2a). The Lgr5-GFP mice are mosaic in adults with no GFP expression in a small part of crypts (Fig. 2b, upper panels). Since Villin-cre is expressed during gut development, it is possible that GFP-silenced patches might be preferentially advantaged due to CREPT deletion. To avoid this possibility, we examined the number of Lgr5-GFP[+] cells in adult mice. Since CREPT was co-localized with Lgr5 and Ki67 (Fig. 2d and Supplementary Fig. 2b) and Lgr5-GFP[+] cells had higher CREPT protein (Fig. 2e) and mRNA (Fig. 2f) level than Lgr5-GFP[−] cells, we deleted CREPT in Lgr5[+] cells of adult *Lgr5-EGFP-ires-creERT2; CREPT[fl/fl]* (Lgr5-CREPT[KO]) mice by addition of Tamoxifen (Supplementary Fig. 2c), and analyzed the Lgr5-GFP[+] cells over time post Tamoxifen treatment (Fig. 2g). The results showed that the deletion of CREPT in Lgr5[+] cells led to a declined number of Lgr5-GFP[+] cells at 7 days post Tamoxifen treatment (Fig. 2h, i). On day 30, the number of Lgr5-GFP[+] cells dropped to a quarter of that at day 0 (Fig. 2, i).

To identify the role of CREPT in stem cells, ex vivo culture of intestinal organoids was generated from the intestinal crypts of WT and Vil-CREPT[KO] mice. However, the CREPT deficient crypts failed to survive ex vivo (Supplementary Fig. 2d). To overcome the failure of organoid formation, we cultured organoids derived from *Lgr5-GFP* and *Lgr5-GFP; CREPT[fl/fl]* mice, and then deleted CREPT by adding 4OH-Tamoxifen (4OHT) supplemented growth medium (Supplementary Fig. 2e). The results showed that the crypts from WT intestines budded at day 3 and formed organoids at day 5 (Supplementary Fig. 2f), however, the 4OH-TAM-induced CREPT deficient crypts produced fewer outgrowth at day 3 and failed to form mini guts at day 5 (Fig. 2j and Supplementary Fig. 2g). Quantification of 100 individual crypts showed CREPT deficient crypts had defective organoid-forming ability ex vivo (Fig. 2k). To evaluate the passage ability of organoids, we harvested the first generation of organoids from crypts and disaggregated organoids to form the second generation. The results showed that WT organoids were re-formed after another 5 days' culture (Supplementary Fig. 2h), but, the Lgr5-CREPT[KO] organoids failed to re-form after disaggregation (Supplementary Fig. 2i). Moreover, we verified the role of CREPT in human colorectal organoids. To deplete CREPT protein, we used a PROTAC, named PRTC, which was developed for specifically targeting CREPT and mediating its degradation[38], to treat human colorectal organoids. The result showed that fewer and smaller organoids formed after PRTC treatment (Supplementary Fig. 2j, k). A Western blot result showed that PRTC induced the decrease of CREPT protein in human organoids (Supplementary Fig. 2l). This result is consistent with the results of organoids formation from mouse crypt cells under CREPT deletion. All these results suggest that CREPT is essential for the maintenance of Lgr5[+] ISCs and organoid formation.

**The intestinal epithelium with CREPT deletion fails to regenerate after damage.** The defects in the intestine and ISCs of Vil-CREPT[KO] mice prompted us to examine whether CREPT plays a role in epithelial regeneration. We applied X-ray irradiation (10 Gy) and allowed the regeneration of intestinal villi and crypts for different times (Fig. 3a). The results showed that all Vil-CREPT[KO] mice reached the euthanasia thresholds within 7 days, but 50% of WT mice remained survival after 14 days post-

irradiation (dpi) (Fig. 3b). IHC analyses demonstrated that X-ray irradiation destructed the crypts, as showed by Olfm4[+] staining, in small intestines of both WT and Vil-CREPT[KO] mice from 0 to 3 dpi (Fig. 3c). At 5 dpi, the crypts (Olfm4[+]) and the villus recovered to normal structure in WT mice, however, the Olfm4[+] crypts barely reformed and the villus structure remained disrupted in Vil-CREPT[KO] mice (Fig. 3c, top panel). A quantitative analysis showed that the numbers of Olfm4[+] crypts were significantly decreased in the Vil-CREPT[KO] mice at 5 dpi (Fig. 3d). Simultaneously, we observed that crypts in the colon were destroyed at 3 dpi but recovered at 5 dpi in WT mice (Fig. 3c, bottom panel), however, the outgrowth of colorectal crypts of irradiated Vil-CREPT[KO] mice was reduced significantly (Fig. 3c bottom panel and 3e). All these results suggest that the intestinal epithelial cells are unable to recover after irradiation when CREPT was deleted.

To validate the results in X-ray irradiation, we challenged the mice with DSS (dextran sodium sulfate), a chemical reagent that destroys the mouse colorectal epithelium and causes inflammation[39], for 7 days, and allowed epithelium recovery (Supplementary Fig. 3a). The results showed that no Vil-CREPT[KO] mouse but 70% of WT mice survived for more than 20 days after DSS treatment (Supplementary Fig. 3b). The architecture of colorectal crypts remained disrupted in Vil-CREPT[KO] mice but was completely recovered in WT mice on day 3 after DSS withdrawal (dpD) (Supplementary Fig. 3c). Taken together, all the results suggest that CREPT is critical for the regeneration of intestinal epithelial cells after destroyed by both physical and chemical reagents.

**CREPT is required for the proliferation and differentiation of Lgr5[+] stem cells.** Next, we explored the function of CREPT in Lgr5[+] stem cells by RNA sequencing (RNA-seq). We induced CREPT deletion in Lgr5-GFP[+] cells by intraperitoneal injection of Tamoxifen into mice, and sorted the Lgr5-GFP[high] cells from disaggregated WT and Lgr5-CREPT[KO] intestines (Fig. 4a). RNA-seq results showed that 198 genes were downregulated by CREPT deletion at homeostasis (Fig. 4b). These genes include differentiation determining genes and cell cycle promoting genes (Fig. 4b, marked genes). Consistently, a Gene Ontology (GO) analysis highlighted that the downregulated genes in CREPT deleted Lgr5[+] cells were enriched in the GO terms of cell differentiation and gene transcription regulation (Fig. 4c). Upregulated genes were those involved in apoptosis (Fig. 4d). In particular, three sets of lineage signature genes, Neurog3, Nkx2-2, Pdx1, and Pax6 (enteroendocrine cell), Spdef and Dll1 (goblet and Paneth cell), and Nox1 and Atoh1 (Paneth cell), were decreased in Lgr5-CREPT[KO] cells (Fig. 4e). Interestingly, CREPT deletion caused downregulation of migration genes (Vim and Snai3), and increased the epithelium-related genes (Lama5 and collagen) (Fig. 4f). This gene expression alteration echoes the observation that EdU-labeled CREPT deleted cells migrated slower along the crypt-villi axis (see Fig. 1). In addition, the expression of cell cycle promoting genes in Lgr5-CREPT[KO] cells were decreased (Fig. 4g) and EdU-labeled crypt numbers in Lgr5-CREPT[KO] organoids were declined (Fig. 4h), suggesting that CREPT is crucial for the proliferation of Lgr5[+] stem cells. Taken together, all the results suggest that CREPT maintains the proliferation and differentiation of Lgr5[+] ISCs at homeostasis.

**CREPT deficiency suppresses the expression of stem cell signature genes in crypts by repressing the proliferation of Lgr5[+] ISCs.** The RNA-seq data of TAM treated Lgr5+ cells showed that the expression of Lgr5[+] ISC signature genes[16], including the Lgr5 gene per se, was barely affected by CREPT deletion in purified

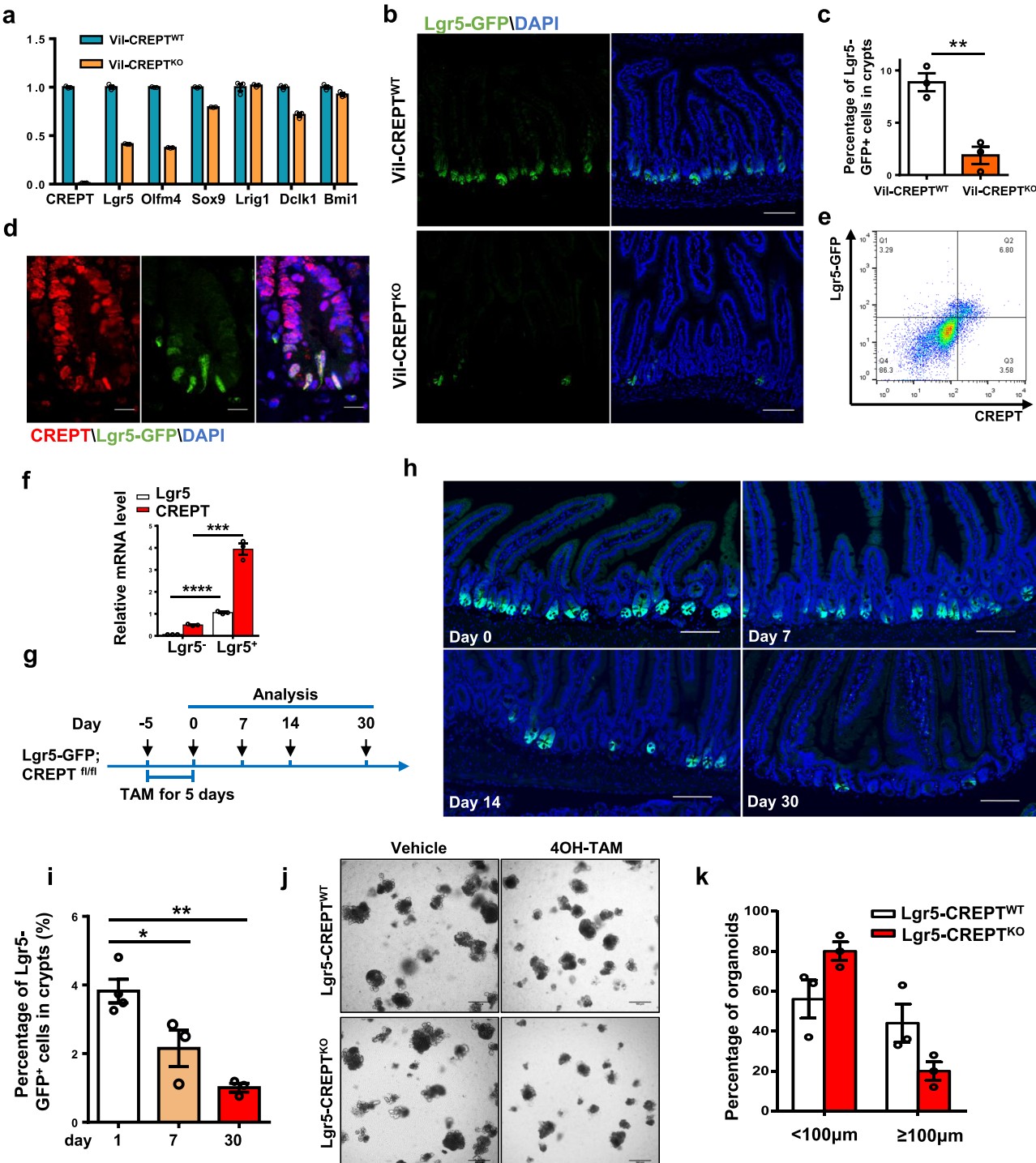

**Fig. 2 CREPT deletion reduces the Lgr5+ cell number. a** A quantitative RT–PCR analysis of the expressions of CBC and reserve stem cell markers in WT and Vil-CREPTKO crypt cells. **b** Representative images of Lgr5-GFP+ cells in WT and Vil-CREPTKO intestines. **c** Quantification of Lgr5-GFP + cells in small intestines of WT and Vil-CREPTKO mice based on FACS analyses. $n = 3$ independent experiments. $p = 0.0042$. **d** Representative images of Lgr5 and CREPT staining in intestine crypts. **e** A FACS analysis of Lgr5 and CREPT stained cell populations. The population of crypts cells, which had the highest level of Lgr5, also expressed the largest amount of CREPT protein. **f** A quantitative RT-PCR analysis of Lgr5 and CREPT in sorted Lgr5-GFP+ and Lgr5-GFP− cells. $n = 3$ independent experiments. $p < 0.0001$ (Lgr5 in Lgr5− vs. Lgr5+ cells). $p = 0.0002$ (CREPT in Lgr5− vs. Lgr5+ cells). **g** A schematic diagram showing the TAM treatment in Lgr5-GFP; CREPTfl/fl mice. **h** Representative images of Lgr5-GFP staining in intestines at indicated time post TAM treatment. **i** Percentage of Lgr5-GFPhigh cells in crypts at indicated time post TAM treatment by FACS analyses. $p = 0.0393$ (day 1 vs. day 7). $p = 0.0011$ (day 1 vs. day 30). **j** Representative images of WT and CREPT deleted organoid formation derived from Lgr5-GFP and Lgr5-GFP; CREPTfl/fl mice. **k** The numbers of WT and Lgr5-CREPTKO organoids showing <100 μm and ≥100 μm diameters. Images are representative of at least three independent experiments. Statistics data represent mean ± SEM. All $p$ values were generated by 2-tailed Student's $t$-test. Scale bars: 100 μm (**b**, **h**, **j**), 10 μm (**d**).

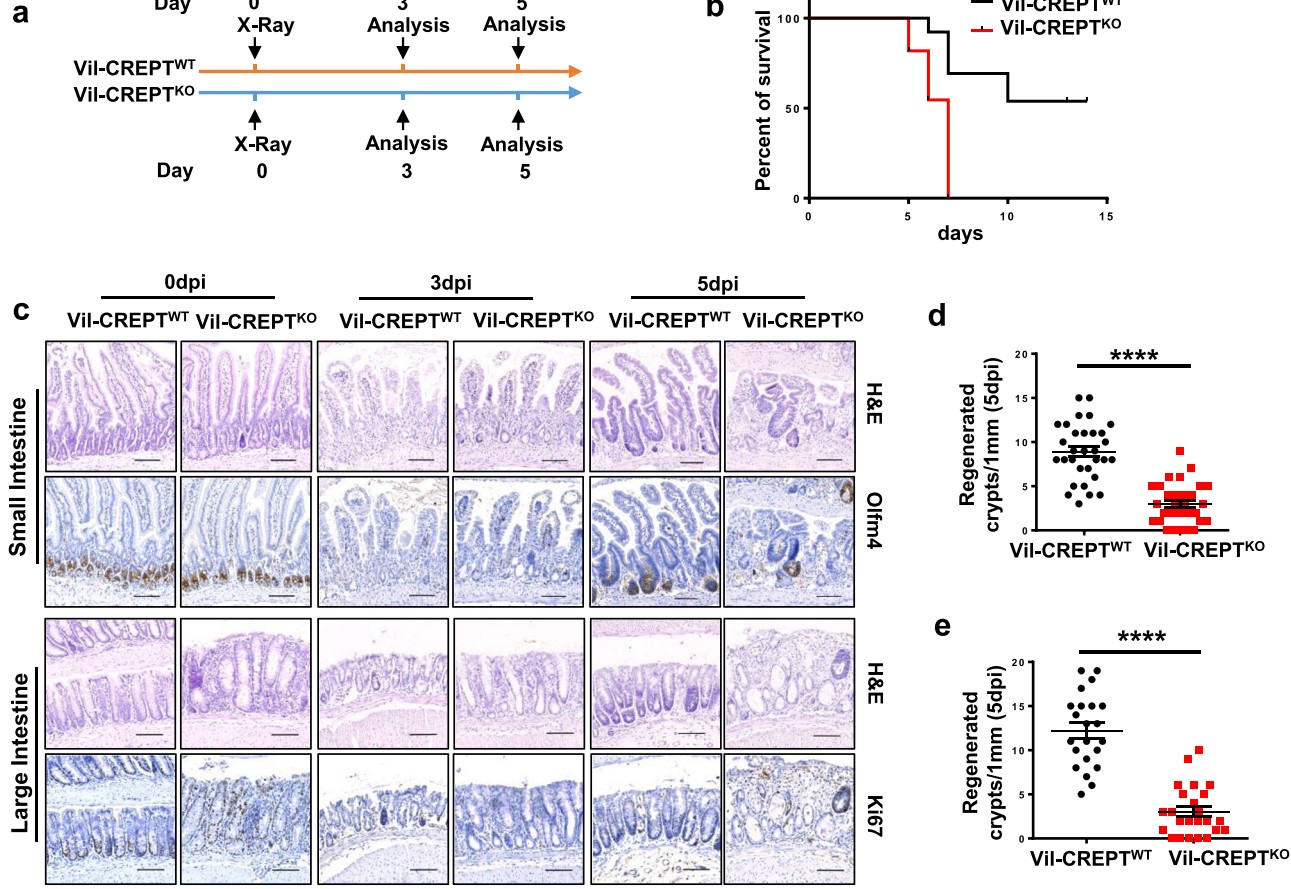

**Fig. 3 CREPT is required for intestinal epithelium regeneration after X-ray irradiation. a** Schematic diagram showing irradiation and analysis schedule. **b** The Kaplan-Meier survival curve of irradiated Vil-CREPT[WT] and Vil-CREPT[KO] mice with $p = 0.0004$ by one-sided log-rank test. $n = 13$ WT mice and $n = 11$ Vil-CREPT[KO] mice. **c** Representative H&E-stained, Olfm4-stained, and Ki67-stained sections of small and large intestines from Vil-CREPT[WT] and Vil-CREPT[KO] mice at different time after 10 Gy X-ray radiation. Images are representative of at least three animals for each condition. **d** Quantification of regenerated crypts per 1 mm small intestine marked by Olfm4 at 5 dpi. $p < 0.0001$ by 2-tailed Student's $t$-test. $n = 3$ independent experiments. **e** Quantification of regenerated crypts per 1 mm colon marked by Ki67 at 5 dpi. $p < 0.0001$ by 2-tailed Student's $t$-test. $n = 3$ independent experiments. Statistics data represent mean ± SEM. Scale bars, 100 μm.

Lgr5+ cells (Supplementary Fig. 4a). However, as mentioned above, we observed a significant decrease of Lgr5-GFP+ cells after TAM-induced CREPT deletion (Fig. 2h, i). It is possible that the cells with ISC signature genes were survived but other cells were lost in CREPT deleted crypts. The RNA-seq data of Lgr5+ cells only reflected the gene expression change in remaining Lgr5+ cells. To clarify the role of CREPT in ISC maintenance, we examined the gene expression in the intestinal crypts from WT and Vil-CREPTKO mice. An RNA-seq result showed that 412 genes were substantially downregulated in the crypts when CREPT was deleted (Supplementary Fig. 4b). We compared our RNA-seq data with a previously identified signature gene set of Lgr5+ stem cells[16]. The results showed that the ISC signature genes were enriched in the downregulated genes in Vil-CREPT[KO] mice, suggesting that CREPT may affect the stem cells in crypts (Supplementary Fig. 4c). For validation, Lgr5 and Olfm4, two stem cell markers, were shown to be decreased in the crypts of Vil-CREPT[KO] mice (Supplementary Fig. 4d). Simultaneously, we observed that cell cycle promoting genes and differentiation determining genes were downregulated (Supplementary Fig. 4e–g). Of note, Nox1 and Atoh1, two marker genes of Paneth cells, were substantially decreased in the crypts of Vil-CREPT[KO] mice (Supplementary Fig. 4f, g), which echoes our immunostaining results for decreased lysozyme positive Paneth cells (Fig. 1j). Taken together, we conclude that CREPT participates in

the regulation of genes in maintaining the proliferation of Lgr5+ ISCs.

**CREPT is required for Wnt activation during the regeneration process.** Next, we analyzed the gene expression profiles of irradiated WT and Vil-CREPT[KO] intestines. Surprisingly, 993 genes were downregulated in irradiated Vil-CREPT[KO] intestines (Fig. 5a), much more than that of untreated WT and Vil-CREPT[KO] intestines (412 genes, Supplementary Fig. 4b). These results suggest that CREPT plays a more prominent role in regeneration rather than homeostasis maintenance. To examine the role of CREPT in intestine regeneration, the RNA-seq data of un-irradiated WT and irradiated WT intestines were analyzed (Supplementary Fig. 5a). A KEGG analysis to identify enriched pathways in intestinal cells after irradiation showed that PI3K-AKT, Hippo, Wnt, MAPK, and TGFβ signaling pathways were upregulated in irradiated WT intestines (Supplementary Fig. 5b). We hypothesized that CREPT deficiency might impede the activation of one or more of these five signaling pathways. Indeed, a gene-set enrichment analysis (GSEA)[40,41] of the gene expression profiles from irradiated WT and Vil-CREPT[KO] intestines showed substantial upregulation and downregulation in Wnt signaling components (Supplementary Fig. 5c), and upregulation mostly in other four signaling pathways (Supplementary Fig. 5d–g). Therefore, we focus on the effect of CREPT on the Wnt signaling pathway.

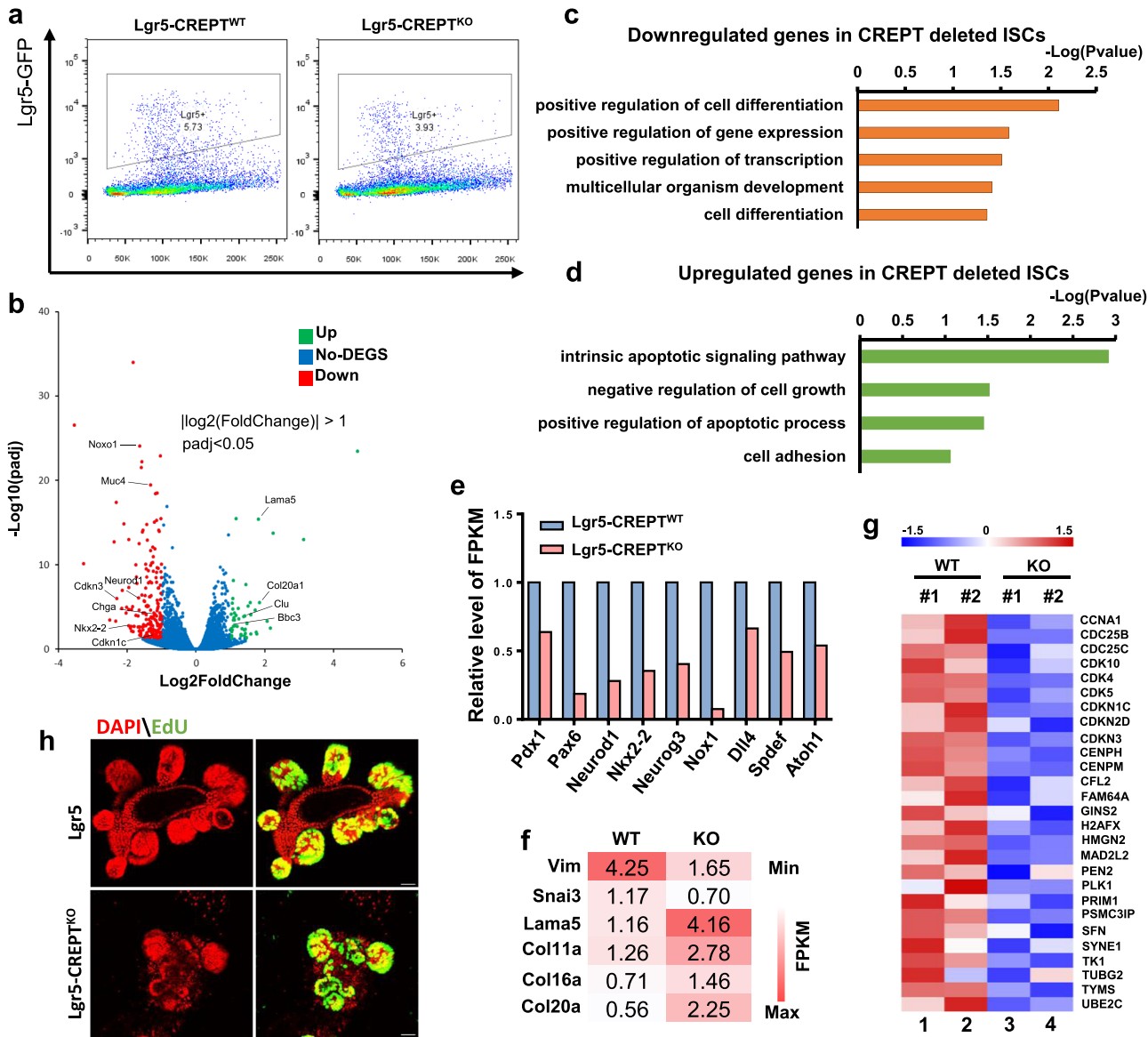

**Fig. 4 CREPT maintains the proliferation and differentiation of Lgr5+ ISCs. a** Gate strategy of sorting Lgr5-GFPhigh cells from TAM treated WT and Lgr5-CREPTKO mice. **b** Volcano plot showing the upregulated and downregulated genes in Lgr5-CREPTKO cells compared to Lgr5-CREPTWT ones. **c, d** GO analysis of downregulated genes (**c**) and upregulated genes (**d**) in CREPT deleted Lgr5+ stem cells, respectively. **e** Relative FPKM showing the expression of differentiation-related genes in WT and Lgr5-CREPTKO cells. Data represent mean of 2 mice per genotype derived from RNA-seq analysis. **f** Heatmap showing the expression of EMT-related genes of WT and Lgr5-CREPTKO cells. $n = 2$ mice for each genotype. **g** Heatmap showing the relative expression level of cell cycle promoting genes of WT and Lgr5-CREPTKO cells. $n = 2$ mice for each genotype. **h** Representative images of EdU-labeled cells in 4OHT treated intestinal organoids derived from WT and Lgr5-CREPTKO mice. Images are representative of at least three independent experiments. Scale bars: 30 μm.

Indeed, CREPT deletion caused substantial downregulation of a majority of Wnt target genes, although *Myc*, *Ccnd1*, *EphB2*, and *Axin2* were unchanged or slightly unexpectedly upregulated (Fig. 5b). We then excluded the Wnt related genes with invariant and/or reduced expression value in irradiated WT mice compared to the untreated WT mice, and analyzed the expression of the increased Wnt responsible genes in the intestines of irradiated WT and Vil-CREPTKO mice. The results showed that many Wnt positive-regulation genes were activated in irradiated WT mice (Fig. 5c, panel 1 vs. 3), but the upregulation was suppressed in irradiated Vil-CREPTKO mice (Fig. 5c, panel 3 vs. 4). Quantitative RT-PCR analysis confirmed that *CD44*, *Ascl2*, *EphB3*, and *Sox9*, four Wnt targeted genes, were substantially increased after X-ray irradiation in WT mice but deletion of CREPT significantly

impaired their expression (Fig. 5d). These results suggest that CREPT is essential for the Wnt signaling activation during the regeneration of intestinal epithelium.

To confirm the regulation of CREPT on Wnt target genes, we performed a ChIP-sequencing (ChIP-seq) experiment in WT and Vil-CREPTKO intestinal crypts. The results demonstrated that CREPT occupancy was enriched at both the promoters and termination regions of *Cd44* and *EphB1*, but only at the termination region of *Alcam* and at the promoter of *Rspo2* (Fig. 5e). The ChIP-seq result was confirmed using the *Cd44* gene by a ChIP experiment (Fig. 5f). These results suggest that CREPT regulates Wnt target genes at the transcriptional level but in different ways. To further confirm the role of CREPT on the regulation of Wnt signal, we used intestinal organoids in the presence of CHIR99021, a GSK3β

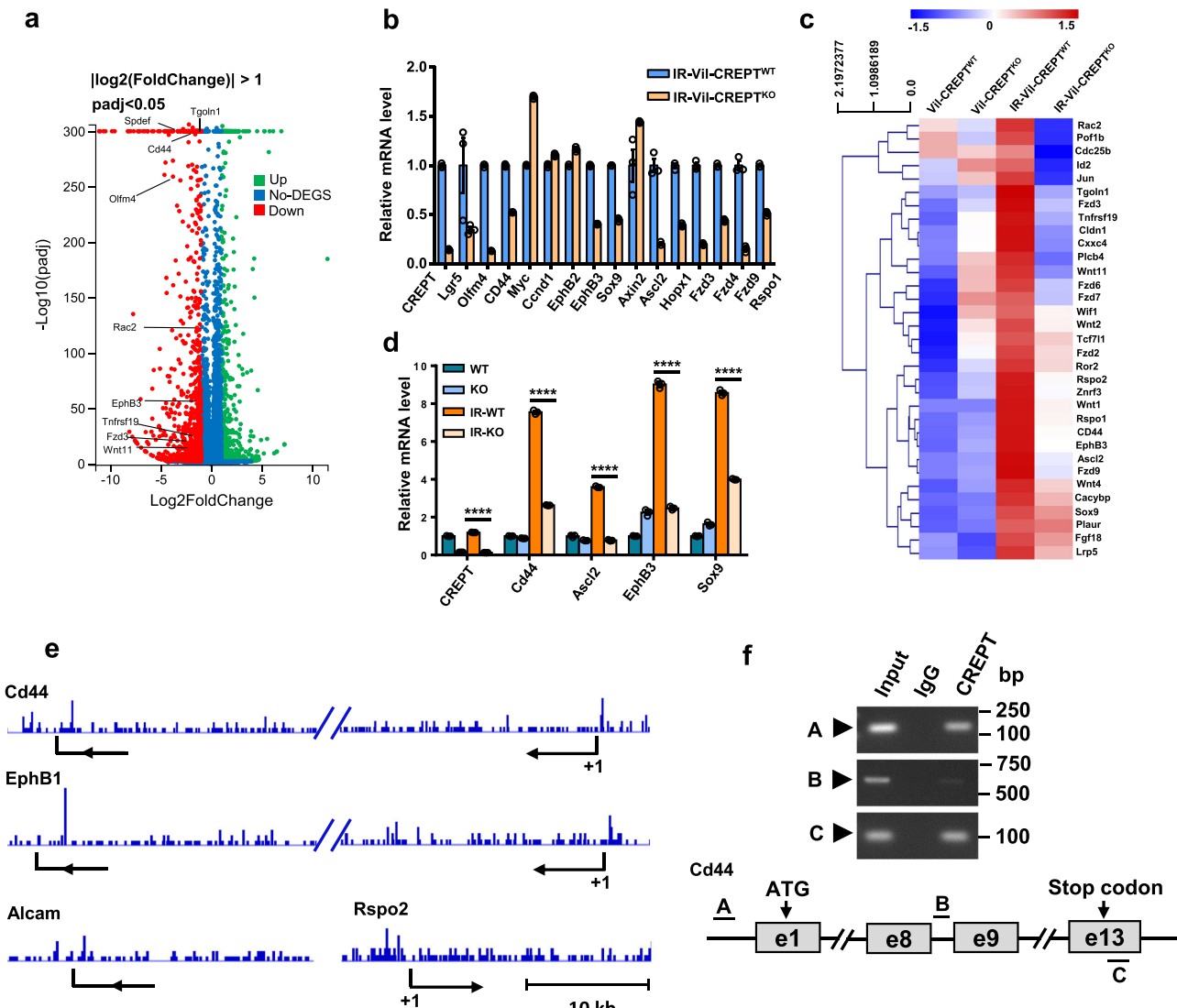

**Fig. 5 CREPT promotes intestinal regeneration through the Wnt signaling pathway.** **a** Volcano plot showing the upregulated and downregulated genes in irradiated (IR)-Vil-CREPT[KO] crypts compared to IR-Vil-CREPT[WT] ones. **b** Quantitative RT-PCR analysis of Wnt target genes in irradiated WT and Vil-CREPT[KO] intestines at 3 dpi. $n = 3$ independent experiments. **c** Heatmap showing the expression levels of Wnt target genes in untreated/irradiated WT and Vil-CREPT[KO] intestines. **d** Quantitative RT-PCR analysis of *Cd44*, *Ascl2*, *EphB3*, and *Sox9* expression in untreated/irradiated WT and Vil-CREPT[KO] intestines. $p < 0.0001$ by 2-tailed Student's *t*-test. $n = 3$ independent experiments. **e** ChIP-seq signals of CREPT occupied genomic loci. The CREPT occupancy frequency is illustrated at the genome of Wnt target genes including *Cd44*, *EphB1*, *Alcam*, and *Rspo2*. **f** Quantitative PCRs showing the enrichment of CREPT on the promotor and termination region of the *Cd44* gene. The primers A, B, and C, were designed as indicated. Images are representative of three independent experiments. Statistics data represent mean ± SEM.

inhibitor to impair the degradation of β-catenin in the cytoplasm[42]. The results showed that CHIR99021 substantially increased the Lgr5[+] staining cells in WT organoids (Supplementary Fig. 5h, comparing β and δ, φ and η), but failed to maintain the organoid passage and Lgr5 expression in the Lgr5-CREPT[KO] organoids (Supplementary Fig. 5h, i, comparing π and η). A quantitative analysis demonstrated substantial decreases in organoid numbers in CREPT KO cells in the presence of CHIR99021 (Supplementary Fig. 5j). These results suggest that CREPT is also essential for the Wnt-activated growth of intestinal organoids.

**CREPT facilitates β-catenin retaining in the crypt nucleus during regeneration**. The impacts of CREPT on the Wnt signaling during intestinal renewal and regeneration prompted us to explore how CREPT regulates Wnt signaling. To this end, we

performed luciferase assays with a widely used super-top repor-ter[43]. The results showed that the deletion of CREPT impaired Wnt, APC-depletion and β-catenin activated transcription of the reporter (Fig. 6a–c) but CHIR99021 greatly rescued the luciferase activity (Fig. 6d), suggesting that CREPT regulates the down-stream events of Wnt signaling at the transcription level. How-ever, the mRNA levels of β-catenin were barely affected by CREPT deficiency (Supplementary Fig. 6). Based on our previous studies[23,28,29], we speculate that CREPT regulates Wnt signaling through β-catenin interaction. Indeed, we observed that endo-genous (Fig. 6e) and exogenous (Fig. 6f) CREPT and β-catenin proteins interacted in the crypts and HEK293T cells, respectively. To decipher the role of CREPT on β-catenin, we examined the protein level of β-catenin. A Western blot showed that the level of β-catenin in the nucleus was substantially decreased in CREPT

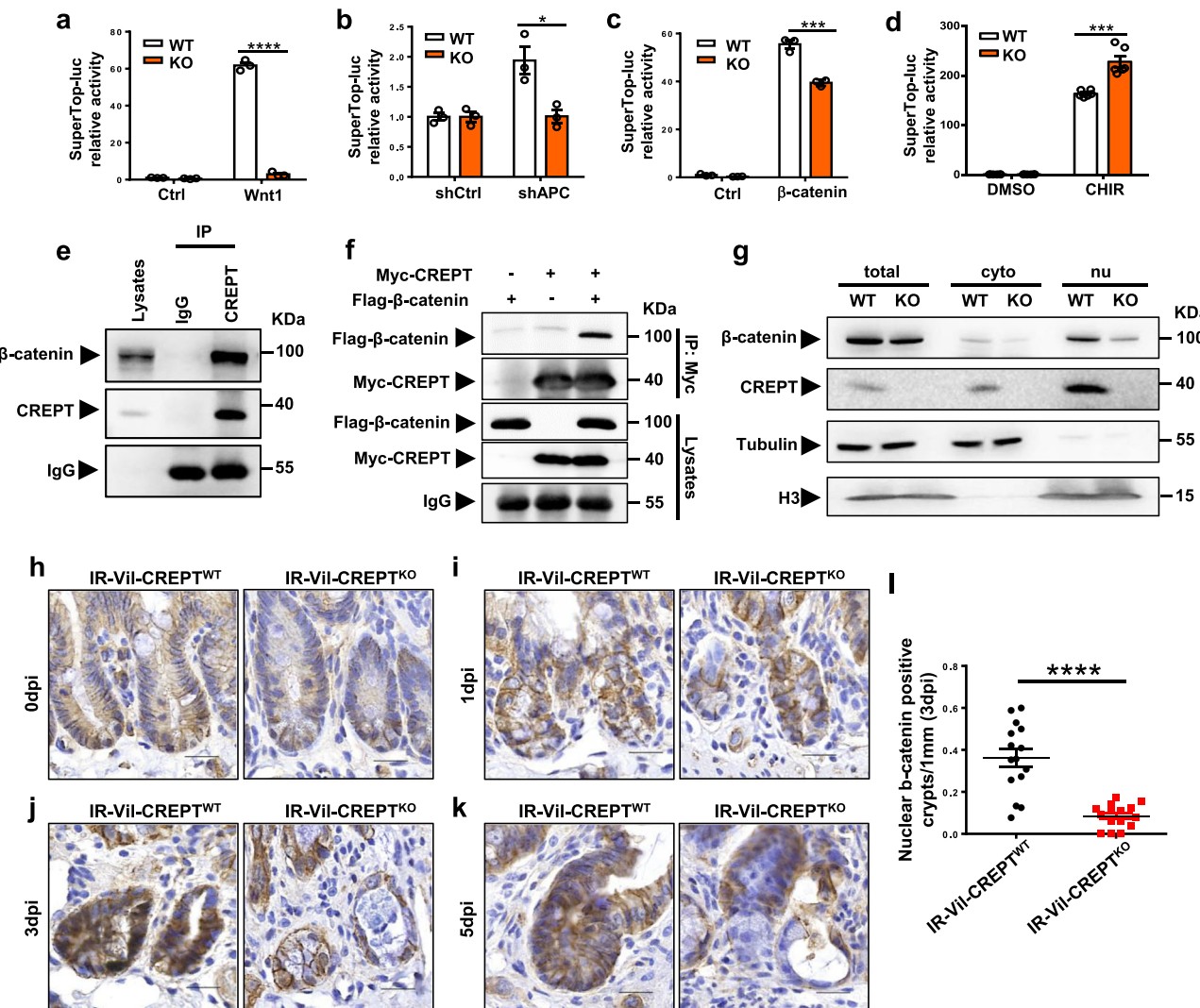

**Fig. 6 CREPT assists β-catenin retaining in the crypt nuclei. a–d** CREPT deletion impaired the SuperTop luciferase activity stimulated by Wnt1 (**a**), APC depletion (**b**), β-catenin (**c**), and CHIR99021 (**d**). CREPT was deleted by a CRISPR-Cas9 system in HEK293T cells. SuperTop-luciferase reporter and pRL-TK plasmids transfected WT and CREPT deleted HEK293T cells were co-transfected with Wnt1 (**a**), shAPC (**b**), and β-catenin (**c**) plasmids, or treated with 5 μM CHIR99021 for 4 h (**d**). The activity was expressed as fold-changes, normalized by an internal control (Renilla). $p < 0.0001$ (**a**). $p = 0.0168$ (**b**). $p = 0.0009$ (**c**). $p = 0.0002$ (**d**). $n = 3$ independent experiments for each treatment. **e** Endogenous CREPT interacts with β-catenin. Cell lysis of intestinal crypts was incubated with control IgG or anti-CREPT antibodies. The immunoprecipitants were analyzed by Western Blotting using anti-β-catenin or anti-CREPT antibodies. **f** Exogenous CREPT interacted with β-catenin. Myc-tagged CREPT (Myc-CREPT) and FLAG-tagged β-catenin (Flag-β-catenin) were co-expressed in HEK293T cells. **g** CREPT deletion reduced the nuclear protein level of β-catenin. CREPT was deleted by the CRISPR-Cas9 system in DLD1 colorectal tumor cells. β-catenin proteins were analyzed by Western blotting in cytoplasmic and nuclear fractions of WT and CREPT KO DLD1 cells. **h–k** Representative images of β-catenin IHC staining in crypts at 0, 1, 3, and 5 dpi after X-ray irradiation, respectively. Obvious nuclear β-catenin appeared at 3 dpi of WT mice, but not Vil-CREPT^KO mice. **l** Quantification of nuclear β-catenin positive crypts per 1 mm intestine at 3 dpi. $p < 0.0001$. $n = 3$ mice per genotype. Images are representative of at least three independent experiments ($n = 3$ mice per genotype). Statistics data represent mean ± SEM. All $p$ values were generated by 2-tailed Student's $t$-test. Scale bars: 20 μm (**h–k**).

KO cells (Fig. 6g). These results suggest that CREPT might regulate the retaining of β-catenin in the nucleus.

To confirm the role of CREPT on retaining β-catenin in the nucleus in vivo, we performed immunohistochemistry experiments. β-catenin was barely observed in the nucleus at 0 and 1 dpi of irradiated WT and Vil-CREPT^KO mice (Fig. 6h, i). Interestingly, overt nuclear β-catenin was detected in the intestinal crypts of irradiated WT mice at 3 dpi (Fig. 6j), and disappeared at 5 dpi (Fig. 6k), suggesting that Wnt signaling was activated at 3 dpi but terminated at 5 dpi during crypt regeneration. However, few cells had β-catenin in the nucleus in the crypts of irradiated Vil-CREPT^KO mice in the whole regeneration processes from 1 to 5

dpi (Fig. 6h–k). A quantitative analysis showed the amount of nuclear β-catenin was significantly reduced when CREPT was deleted at 3 dpi (Fig. 6l). All these results suggest that CREPT regulates the nuclear retaining of β-catenin in crypt cells during intestinal regeneration.

## Discussion
The ISCs are precisely controlled to enable intestinal daily renewal and regeneration after damage, yet the regulating mechanism is not fully understood. In this study, we report that CREPT is one of the key regulators of ISCs. We observed that CREPT was mainly expressed in crypt cells and CREPT

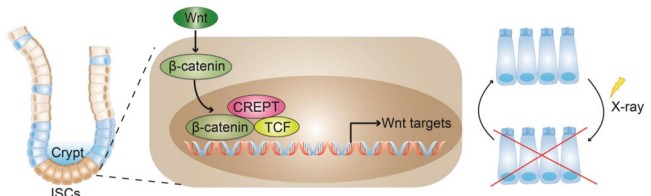

**Fig. 7 A graphical model of CREPT function in ISCs and intestinal regeneration.** ISCs reside in intestinal crypts. CREPT maintains the proliferation and differentiation of Lgr5+ ISCs at homeostasis. Under damages such as irradiation, CREPT promotes the activation of the Wnt signaling pathway through facilitating β-catenin retention in the nucleus. CREPT is indispensable for intestinal regeneration, and the deletion of CREPT leads to impaired proliferation ability of ISCs and failure of intestinal regeneration.

deficiency decelerated the fast turnover of intestinal epithelia and impacted the differentiated cell number. In particular, a great loss of Lgr5+ stem cells were found in CREPT deleted crypts. In addition, the CREPT-deleted intestinal epithelia were not able to regenerate after irradiation and DSS treatment. All these observations suggest that CREPT plays a critical role in the maintenance and proliferation of ISCs. Mechanically, we identified that CREPT regulated the proliferation and differentiation of Lgr5+ ISCs at homeostasis, and participated in the Wnt signaling activation by retaining β-catenin in the nucleus in ISCs during intestinal regeneration. Taken together, our results suggest that CREPT functions to maintain ISCs in intestines (Fig. 7).

A previous study showed that ablation of Lgr5+ cells was tolerated by the intestine, but led to epithelia collapse after irradiation[17]. This echoes our observation that Vil-CREPT^KO mice survived but irradiated Vil-CREPT^KO intestine failed to regenerate (Fig. 3 and Supplementary Fig. 3). Together with our findings that Lgr5+ cell number was significantly reduced in Vil-CREPT^KO intestine, we conclude that the regeneration failure of CREPT deficiency intestine might be due to a reduced number of Lgr5+ cells. In this context, the expression of CREPT should be prior to Lgr5 during embryonic development. Indeed, CREPT is abundantly expressed in the early embryo before E13.5[23] and Lgr5 is expressed at E16.5[44]. On the other hand, we noticed that the expression of CREPT in the crypts was extended to the TA cell zone, much broader than the expression of Lgr5 (see Fig. 1). This fact suggests that CREPT is not a stem cell marker specifically expressed in ISCs, although it plays a critical role in the proliferation of ISCs. Of note, the Lgr5-cre driven CREPT deletion showed no significant phenotype under irradiation treatment (Supplementary Fig. 7) while the Vil-CREPT^KO mice were very sensitive to irradiation (see Fig. 3). These observations suggest that CREPT affects the stem cell during the early gut development, although we could not exclude the possibility that Lgr5-cre is mosaic in adult mice with no cre expression in a part of Lgr5+ cells (see Lgr5-GFP staining in untreated WT mice in Fig. 2b, h). To distinguish the roles of CREPT between intestinal development vs. in adult tissue homeostasis, the inducible Villin-cre or Ah-Cre[45] mice should be used to delete CREPT in adult intestines.

We noted that the Villin-cre driven CREPT knockout phenotype was different between in vivo vs. in vitro. The intestinal epithelium formed normally in vivo, although the quantity of Lgr5+ stem cells and differentiated cells were reduced in CREPT KO mice (Figs. 1j, k, 2b, c). However, in the purified epithelial cell culture system, the crypts from CREPT KO mice could not form organoids with EGF, Noggin and R-spondin1 supplements (supplementary Fig. 2d). We speculate that the phenotypic difference between in vivo and in vitro was due to the different

niches the epithelial cells reside in. It has been known that Wnts could be produced by Paneth cells and stomal cells[7,46,47]. Under in vivo conditions, although the quantity of Paneth cells was reduced in the crypts of Vil-CREPT^KO intestines, stromal cells could produce enough Wnt ligands to maintain the stem cell proliferation[46]. However, under the in vitro condition, due to lack of stromal cells, reduced Paneth cells in the crypt base could not secrete enough Wnts, leading to the failure of organoid formation.

Although our previous results demonstrated that CREPT is required for the Wnt signaling activation in tumor cells[28], we noticed that not all the Wnt downstream genes responded to CREPT deletion in ISCs. Several canonical Wnt target genes, such as *CCND1* and *c-Myc*, were not repressed, but slightly elevated in irradiated Vil-CREPT^KO mice. This surprised us because CREPT was shown to elevate *CCND1* and *c-Myc* expression significantly through Wnt signaling in tumor cells[23,28]. We reason that the role of CREPT on Wnt activation in normal intestine regeneration might be different from that during tumorigenesis. Consistent with our observations, other studies showed that *Cyclin D1* null mice failed to exhibit any abnormalities in the intestine[48] and *APC* deletion in intestines had no effect on the immediate *Cyclin D1* expression[49,50]. Also, *c-Myc* was found dispensable for intestine homeostasis[51]. On the other hand, the unexpected regulations of Cyclin D1 and c-Myc by CREPT deletion might be due to a complementary effect during tissue regeneration. This hypothesis is supported by the essential role of c-Myc for intestinal proliferation[52,53]. In this context, the upregulation of cyclin D1 and c-Myc might be regulated by other signal pathways, rather than the Wnt pathway. To support this notion, our GSEA analysis showed major upregulation in PI3K-AKT, Hippo, MAPK and TGFβ signaling pathways in Vil-CREPT^KO intestines during regeneration (Supplementary Fig. 4d–g). Therefore, we postulate that cyclin D1 and c-Myc are upregulated through these signaling pathways rather than Wnt signaling during regeneration, but the upregulation of cyclin D1 and c-Myc could not compensate for the loss of CREPT and other Wnt target genes during intestinal regeneration. Furthermore, it is quite interesting that CREPT occupancy occurs differently at different Wnt target genes (Fig. 5e). In *Cd44* and *EphB1*, CREPT occupied both the promoter and termination region. This two-location occupancy implies a transcriptional loop, which was observed in tumor cells in our previous study[23]. However, CREPT also occupies solely either the promoter or the termination region in *Rspo2* or *Alcam*. These results suggest that CREPT might regulate different genes in coordination with other factors. At homeostasis, we could barely observe any nuclear β-catenin in both WT and Vil-CREPT^KO intestinal crypts, but CREPT deletion downregulated the expression of many stem cell signature genes, lineage marker genes and cell-cycle promoting genes, which are mainly regulated by Wnt signal[21,46,54,55]. Taken together, we speculate that Wnt signaling regulates different genes expression through CREPT under different physiological conditions. However, future work is needed to prove this hypothesis.

We found that CREPT facilitated nuclear β-catenin retention in crypt cells. This may represent one of the ways how CREPT promotes Wnt signaling in the intestinal stem cell zone. However, the mechanism as to how CREPT retains β-catenin in nuclei is not fully understood. We speculate that CREPT, which is a nuclear protein, prevents β-catenin from nuclear export by interacting with β-catenin and recruiting other β-catenin nuclear binding proteins, such as BCL9[56]. Still, it remains unclear whether CREPT regulates β-catenin/TCF transcription activity during regeneration. We hope this will be done in our future studies.

One group reported that ISCs survived irradiation because these cells were more capable of DNA repair than the

differentiated cells[57]. The DNA repair in ISCs utilizes homologous recombination (HR), with fewer repair mistakes than non-homologous end joining (NHEJ)[57]. Coincidentally, CREPT has been reported to be involved in the DNA repair complex to promote DNA double strain breaks (DSBs) repair by both HR and NHEJ[32,58,59]. In this study, we provided evidence that CREPT participated in the regulation of intestine regeneration after irradiation and chemical-induced damage. Although we obtained strong evidence that CREPT activated Wnt signaling for ISC proliferation, we could not exclude the possibility that CREPT might promote DNA repair in ISCs after irradiation. We believe that CREPT might also function on DNA damage repairing for the regeneration of intestines. This required more effort to elucidate the detailed mechanism in further studies.

## Methods

**Mice**. Vil-cre mice[33] and Lgr5-EGFP-IRES-creERT2[2] mice were kindly provided by Prof. Ye-guang Chen of Tsinghua University. These mice lines were maintained in a C57BL/6 genetic background. Both male and female mice were used in the study. All mice were housed in isolated ventilated cages (maximally six mice per cage) in a barrier facility at Tsinghua University. The mice were maintained on a 12-h light-dark cycle, 22–26 degrees with sterile pellet food and water ad libitum. The laboratory animal facility has been accredited by AAALAC (Association for Assessment and Accreditation of Laboratory Animal Care International) and the IACUC (Institutional Animal Care and Use Committee) of Tsinghua University approved all animal protocols used in this study.

For EdU pulse-chase experiment, the WT and Vil-CREPTKO mice were treated with a single EdU injection (100 μg) and analyzed after different times. For the cre induction by Tamoxifen, the Lgr5-GFP-CreERT2 mice (as WT group) and Lgr5-GFP; CREPTfl/fl mice (as experimental group) both were intraperitoneally injected with tamoxifen at 1 mg a day for 5 consecutive days. Then, Lgr5-WT and Lgr5-CREPTKO cells were harvested by sorting Lgr5high cells on day 7 for RNA-seq, and intestinal sections were analyzed at different times post treatment.

For irradiation, mice received a single dose of abdominal X-ray radiation (10 Gy) and were then analyzed at different time points. For DSS treatment, WT and Vil-CREPTKO mice were fed with 2.5% DSS in water for 7 days. We observed the behavior and health condition of irradiated/DSS-treat mice every day. The mice are judged to be dead and euthanized by CO2, when the euthanasia thresholds meet the following predefined criteria. (1) The animal is on the verge of death or unable to move, or has no response after giving gentle stimulation; (2) Difficulty in breathing: typical symptoms are saliva and/or cyanosis; (3) Diarrhea or incontinence; (4) Reduced body weight by 20% before the experiment; (5) The animal is unable to eat or drink; (6) The animal has obvious anxiety and restlessness; (7) Paralysis, persistent epilepsy or stereotyped behavior, etc.; (8) Animal skin damage area accounts for more than 30% of the whole body, or infection and purulent appear; (9) Other circumstances where the veterinarian determines that an endpoint is required.

**Isolation of murine intestinal epithelial cells and organoid culture**. Murine small intestines were harvested, and cut open longitudinally. After washing in cold PBS with Penicillin (200 U/ml) + Streptomycin (200 μg/ml) (Invitrogen) for 3 times, the intestines were cut into 5 mm pieces, and incubated in pre-cold PBS with 2 mM EDTA on ice for 1 h. The intestines were then transferred to new cold PBS. Samples were vigorously shaken again for 2 min, and the intestinal epithelial cells (villi + crypts) were enriched in the supernatant. The supernatant was either centrifuged to collect the whole epithelial cells or passed through 70 μm cell strainers (Corning) to filter out villi. The crypts were spun down as centrifuged at 300 × g for 5 min. The crypts were embedded in 50 μl Matrigel (Corning) and seeded in pre-wormed 24-well plates. Then, the crypts in Matrigel were cultured in DMEM/F12 medium (Invitrogen) containing 50 ng ml$^{-1}$ EGF (Peprotech), 100 ng ml$^{-1}$ Noggin (Peprotech), 500 ng ml$^{-1}$ R-spondin1 (kindly provided by Prof. Xinquan Wang of Tsinghua University), N-acetylcysteine (Sigma-Aldrich), N2 and B27 (Invitrogen).

**Human colorectal organoid culture and PRTC treatment**. The human colonic mucosa sample was obtained trimming surgically resected specimens under informed consent and institutional review board approval at the Peking University People's Hospital. The human colorectal stem cells were obtained as described previously[60]. Briefly, the resected specimens were washed and cut into 5 mm pieces. The dissected samples were incubated in pre-cold PBS with 2.5 mM EDTA, placed on a rocking shaker and rocked gently at 4 °C for 1 h. The samples were then transferred to new cold PBS, and pipette up and down to disaggregate the crypts. The crypts were spun down at 300 × g for 5 min and re-suspended in Matrigel. The cells were cultured in Advanced DMEM/F12 medium (Invitrogen) containing 50 ng ml$^{-1}$ EGF, 100 ng ml$^{-1}$ Noggin, 500 ng ml$^{-1}$ R-spondin1, 100 ng ml$^{-1}$

Wnt3A (R&D), N-acetylcysteine, B27, 500 nM A83-01, 10 μM SB202190, and 10 nM Gastrin I (R&D).

The CREPT targeting PROTAC (PRTC) was synthesized as described previously[38]. Briefly, the target arm of PRTC was designed according to the lysine 266 to valine 286 of CREPT, connecting with 6-aminohexanoic acid (AHX) and a VHL ligand. Mutant PRTC (leucine 269, 276, and 283 were replaced with proline residues) failed to interact with CREPT and was added to the culture medium as control. The human colorectal organoids were treated with PRTC and mutant PRTC (Ctrl), respectively, at 10 μM for 5 consecutive days.

**Immunoblotting and immunoprecipitation**. For endogenous interaction assays, protein lysates were prepared from intestinal crypts in cell lysis buffer (50 mm Tris-Cl, 150 mm NaCl, 1% Nonidet P-40, 0.5% sodium deoxycholate, and 1% SDS, pH 8.0) with protease inhibitors. For exogenous interaction assays, Myc-CREPT and Flag-β-catenin were co-transfected into HEK293T cells. After 24 h of transfection, the cells were lysed in lysis buffer. The lysates were incubated with appropriate antibodies at 4 °C overnight, followed by the addition of protein G-agarose beads to pellet the immune complexes for 1 h. Cell lysates and immunoprecipitants were analyzed by immunoblotting for the indicated proteins. Original uncropped immunoblots are shown in Supplementary Fig. 8.

**Immunohistochemistry and immunofluorescence**. Mouse tissues were washed three times with cold PBS, fixed in 4% paraformaldehyde solution, and embedded in paraffin. 5 μm sections were de-paraffined by xylene and hydrated in graded alcohol. Antigen retrieval of tissue sections was performed by sodium citrate buffer and quenched by a peroxidase-blocking solution (Dako). Sections were then incubated in protein block solution (Dako) for 10 min, and overnight at 4degrees with primary antibodies, including mouse anti-CREPT[61], rabbit anti-ki67 (Abcam, ab15580, 1:500), rabbit anti-Olfm4 (CST, 39141, 1:400), rabbit anti-Lysozyme (Dako, A0099, 1:1000), mouse anti-β-catenin (BD, 610154, 1:200), rabbit anti-cyclin D1 (Abcam, ab21699, 1:100), rabbit anti-GFP (CST, 2956, 1:200).

For immunohistochemistry, HRP rabbit/mouse secondary antibody solution (Dako) was added on sections after primary antibody, and color developed by DAB (Dako). The blue nucleus was stained by hematoxylin solution (Sigma, 03971). Goblet cells were stained by alcian blue kit (Abcam, ab150661) according to manufacturers' recommendations. Slides were then dehydrated by xylene, and mounted by Leica cv ultra medium. Bright-field pictures of sections were taken by the Zeiss Axio Scan.Z1 slice scanner.

For immunofluorescence, Alexa Fluor 488 conjugated anti-rabbit (CST, 4412, 1:1000) and 546 conjugated anti-mouse (Thermo, A11030, 1:500) secondary antibody were added to the slides after the primary antibody, and DAPI (Sigma, D8417) was used to detect the nucleus. Slides were mounted by ProLong™ Gold anti-fade reagent (Thermo, P36934), and the fluorescent pictures were taken by the FV1200 laser scanning microscope (Olympus).

For organoids staining, the organoids were fixed overnight in 4% formaldehyde solution at 4degrees, permeabilized in 0.3% Triton for 20 min, and blocked in 2% BSA for 1 h at room temperature. Then, the organoids were incubated overnight with primary antibodies at 4degrees. After washed in PBST (0.1% Tween 20 in PBS) for 3 times, the organoids were incubated in fluorescent antibodies overnight at 4degrees. The DAPI was applied to detect the nucleus. For the EdU experiment on organoids, the intestinal organoids were incubated in culture media supplemented with 10 μm EdU for 1 h. EdU positive cells were detected by Click-iT™ EdU Alexa Fluor™ 488 Assay Kit (Thermo, C10425) according to manufacturers' recommendations.

**Isolation of Lgr5-GFP⁺ cells**. Crypts were collected from Lgr5-WT and Lgr5-CREPTKO mice as the method described above, and suspended in TryPLE (Gibco, 12604) to be disaggregated into single cells. The disaggregated cells were passed through 40 μm cell strainers (Corning) and analyzed by flow cytometry (BD Arial III) based on the fluorescent signals of GFP proteins.

**RNA-seq**. RNAs were extracted from crypts of WT and Vil-CREPTKO mice and/or the intestines of X-ray irradiated WT and Vil-CREPTKO mice using Trizol (Invitrogen) according to manufacturers' recommendations. High-throughput sequencing was performed by BGISEQ (BGI genomics). The RNA-seq was carried out with two biological replicates. The RNA-seq results were mapped to the GCF_000001635.26_GRCm38.p6 genome sequence from NCBI. Clean reads were mapped to reference transcripts using Bowtie2, and then calculate the gene expression level for each sample with RSEM. DEseq2 algorithms were used to identify the Differentially expression genes. Genes with absolute log2-transformed fold changes >1 were regarded as differentially expressed genes, and a threshold of q value <0.05 was used. Gene Ontology (GO) analysis was performed with DAVID Bioinformatics tools (https://david.ncifcrf.gov/). The full ordering lists of GO terms enriched in Fig. 4 are shown in the Source Data file. Heatmaps were generated by MeV (http://mev.tm4.org/) and Gene set enrichment analyses (GSEA) were performed with the GSEA platform of the Broad Institute (http://software.broadinstitute.org/gsea/index.jsp) using the pre-ranked lists of gene fold changes against public gene sets of intestinal gene signature.

**Quantitative RT-qPCR**. After RNA extraction, cDNA was prepared from 2 µg total RNA using the Fast King reverse transcription kit (Tiangen). The PCR reactions were performed on LightCycler 480 (Roche) as indicated by the manufacturer. The relative transcript level of each gene was normalized to β-actin, and calculated according to the $2^{-\Delta\Delta Ct}$ method. The primers used were listed in Supplementary Data 1. Original uncropped PCR results are shown in Supplementary Fig. 9.

**ChIP and ChIP-seq**. All ChIP assays were performed in intestinal crypt cells. The crypts were isolated as the method described previously, and then suspended in TryPLE (Gibco, 12604) to be disaggregated into single cells. The cells were crosslinked with 1% formaldehyde for 10 min at 37 °C, and the nuclei were extracted according to the manufacturer's instructions (CST, 9004). The DNA was sheared to a length of approximately 100–500 bp by sonicating the samples with 20 pulses. Samples were incubated on ice between each pulse. ChIP assays were then carried out with the manufacturer's instructions, using the anti-CREPT antibody. For ChIP–quantitative PCR, the immunoprecipitated DNA or input DNA was analyzed by PCR, and the primers used were available in Supplementary Data 1. For ChIP-seq, CREPT immunoprecipitated DNA was ligated with adapters, amplified by PCR, and sequenced by BGISEQ (BGI genomics). Clean reads were mapped to the GCF_000001635.26_GRCm38.p6 genome sequence from NCBI using SOAPaligner/soap2[62]. Peak (ChIP Sequencing enrichment area) scanning in the whole genome was performed to obtain the position information of Peak on the genome and the sequence information of the Peak region by MACS[63]. Heatmaps of CREPT enrichment regions were visualized by using IGV.

**Nuclear protein extraction**. WT and CREPT knockout DLD1 cells were suspended in 600 µl of ice-cold Buffer I (10 mM HEPES, 1.5 mM $MgCl_2$, 10 mM KCl, and protease inhibitors, pH 8.0) and incubated on ice for 15 min. NonidetP-40 was then added to a final concentration of 1%. Samples were shortly vortexed and centrifuged at top speed for 2–3 min. The supernatants were collected as the cytoplasmic fraction. The nuclear pellets were re-suspended in 220 µl of ice-cold Buffer II (20 mM HEPES, 1.5 mM MgCl2, 420 mM NaCl, 0.2 mM EDTA, 25% glycerol, and protease inhibitors, pH 8.0) and then rotated vigorously for 30 min. Samples were centrifuged at top speed for 10 min, and the supernatants were collected as the nuclear fraction. All steps were performed at 4 °C.

**Statistics**. All experiments reported in this study were repeated at least three independent times. In vivo analyses were conducted with at least three animals per condition. The RNA-seq was carried out with two biological replicates. The number of animals in each experiment is depicted in Figure legends. Intestinal organoids of each group were cultured in 3 wells repeatedly, the organoids with ≥100 µm diameters were considered. GraphPad Prism version 6 was used for statistical analysis. Two-tailed unpaired Student's $t$-tests were used for statistical significance analysis of two groups to generate $P$ values.

**Reporting summary**. Further information on research design is available in the Nature Research Reporting Summary linked to this article.

## Data availability

The authors declare that all data supporting the findings of this study are available upon reasonable request. The RNA-sequencing data reported in this study have been deposited in the Gene Expression Omnibus (GEO) database under accession codes: GSE143695, GSE143604 and GSE143605. The ChIP-seq data have been deposited in the GEO database under accession code: GSE158243. Source data are provided with this paper.

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

## Acknowledgements
We thank Prof. Ye-guang Chen of Tsinghua University for kindly providing *Vil-cre* mice and *Lgr5-EGFP-IRES-creERT2* mice. We thank Prof. Xinquan Wang of Tsinghua University for kindly providing R-spondin1. We thank Prof. Xia Wang of Tsinghua University for kindly providing help in building up the human organoid culture system. We thank Dr. Hans Clevers of Hubrecht Institute for his support and help in this project initiation. We thank the State Key Laboratory of Membrane Biology and Center of Biomedical Analysis at Tsinghua University for providing confocal microscopy, slice scanner, and flow cytometry. We thank the Laboratory Animal Research Center of Tsinghua University for animal maintenance. This work was supported by grants from the National Natural Science Foundation of China (81872244; 81830092; 81572728; 81572729; 81872249), and the Chinese National Major Scientific Research Program (2016YFA0500301).

## Author contributions
L.Y., Y.W., and Z.C. conceived the study. L.Y., X.Y., and Z.C. wrote the paper. L.Y. performed the experiments. H.Y and Y.S. contributed to experimental design and performances. L.D., B.Z., and W.Z. created and characterized CRISPR-Cas9 mediated CREPT deleted cell lines. X.W. and Y.K. contributed to RNA-seq data analysis. F.R. and W.W. contributed to experimental design. X.Y., Y.C., and B.J. contributed to histology analysis. Y.S. created the graphical model in Fig. 7.

## Competing interests
The authors declare no competing interests.
