## [Peer Review File · Nature Communications]

Reviewers' Comments:

Reviewer #1:

Remarks to the Author:

The manuscript by Chang and colleagues explores the role of a candidate cancer driver gene, CREPT, in maintaining intestinal stem (Lgr5+) cells (ISCs) and in controlling intestinal regeneration. Using intestinal specific deletion of CREPT the authors describe the resultant phenotypes, finding modest effects on the steady state and rather more dramatic effects when the gut is forced to enter a regenerative phase by high dose irradiation. The initial motivation seems to have been to document the physiological role of CREPT in a whole tissue/animal setting because this has not yet been done. The authors find that stem cells as defined by a number of markers but mainly Lgr5 are underrepresented in adult intestinal epithelium following deletion of CREPT in the developing gut due to Cre-mediated deletion (VilCre model).

It is worth noting that CREPT is viewed as a cell cycle regulator that regulates directly (promotes binding of RNA pol II) or enhances transcription of a number of cell cycle regulatory genes. In this respect it is somewhat surprising that there is not more attention paid to the cell cycle related changes occurring in the intestinal epithelium. Is replication stalled; if so at what stage and are damage markers induced?

The authors present a significant amount of data that in the main seems to have performed to a good standard albeit with some missing and important technical details. Their conclusion about ISCs being maintained by CREPT is correct but how and when that maintenance is achieved remains somewhat elusive. The phenotypic characterization of steady and regenerative states seems good. Mechanistic insights are relatively superficial: molecular profiling tools are used almost exclusively and it is often unclear where whether they are describing the phenotype or elucidating it.

In the steady state CREPT KO mice show grossly normal small intestine but enlarged crypt epithelium in the colon. In both tissues proliferation is compromised and secretory and ISCs are underrepresented. In an apparent adaptation cellular migration of cells along the crypt to villus axis is slowed. Of note the interpretation of reduced ISCs numbers depends on cell counts detecting GFP expressed from the Lgr5-EGFP-iresCreERT2 mouse line made by Barker but with CREPT still deleted due to VilCre. The Lgr5 mice are notoriously mosaic in adults with some crypts expressing the transgene and others not. Given that the Villin is expressed during gut development it is possible that silenced patches are preferentially advantaged following deletion of CREPT. Some reassurance is needed in this regard before fully accepting the data shown in Fig2.

Minor point: the IF for CREPT shown in Fig1A or Suppl Fig1 C does not convince that there it is nuclear localized and seems to show strong apical staining throughout the crypt. These images contrast with Fig2D where nuclear staining is obviously present.

The sensitivity of the CREPT deficient mice to irradiation compared to WT seems clear. One point of concern is the wording describing how survival analyses were done (page 7) where mice are said to have died due to the irradiation. From an animal ethics standpoint mice receiving such treatments should be on enhanced monitoring schemes and culled when reaching a predefined threshold of departure from health. Reassurance on this point is required.

Minor point: The time post IR for the counts shown in Fig3 D and E should be stated.

The data in Fig4 and 5A describes transcriptional profiling of purified ISCs isolated following deletion of CREPT using the Lgr5CreERT2 model. Missing from the experimental description is the time post tamoxifen treatment when tissues were harvested (the mice receive 1mg a day for 5 days) and if the controls also received tamoxifen. This analysis is welcome and informative but some phenotypic analysis over time would have enriched it. (Do the number of Lgr5+ cells

decrease with time following CREPT deletion?). The analysis shows minimal impact on most ISC associated regulatory genes but significant changes in transcriptional abundance of cell cycle associated genes.

The analysis of Gene Ontology terms lacks detail: were the 5 GO terms for downregulated genes and 4 for upregulated gene the top 5 and 4 that came from the analysis? If not the full ordering with their significance should be shown perhaps in supplementals. There are a lot of changes in cell cycle associated genes. Is it not surprising that these did not generate a significant grouping in the GO analysis?

The data and related text describing Fig5 is rather odd. This is an analysis of similar markers to those shown in Fig4 but now looking at whole crypt epithelium. There is considerable duplication with elements of Fig 1 and 2 in demonstrating that ISCs and secretory cells are under represented in the CREPT *VilCre* knockouts. The authors seem to accept this but to also refer to CREPT "suppressing expression" of these ISC genes. Yet Fig4 seems to make clear that these genes are largely not so regulated. The reduction in transcriptional abundance in the whole crypt analysis would seem merely to reflect the relative abundance of the cells present.

A striking feature of comparing Fig4G and Fig5E is that changes in cell cycle genes are near identical in both ISC and whole crypts. It is difficult not to feel that these effects are the predominant consequences of CREPT deletion, more so than the effects on ISCs.

The final sections of the manuscript search for mechanistic insight by probing the differential changes in transcriptional abundance in WT and CREPT deficient gut following irradiation. They identify that five trophic pathways (by GO term analysis) are implicated in the regenerative response in both genotypes. (Again are these the top 5 terms obtained from the analysis or were they selected?) Operating on the hypothesis that CREPT deficiency might impede activation of one or more of these pathways the authors look for a confused response where genes that are upregulated in WT are downregulated in the knockout. They identify genes associated with Wnt signaling as showing this behavior. However it is not entirely clear what feature of the gene set enrichment analysis allows this selection as PI3K-AKT, Hippo and MAPK similarly have sets of genes behaving this way. Is the normalized enrichment score for Wnt really different from these others or are they all much the same? Are negative regulators behaving differently from positive regulators?

Again one wonders the extent to which the analyses shown in Figure 6 is describing the phenotype rather than identifying the mediators of it. With that in mind some definitive and/or orthogonal validation is important. Figure 7/Suppl Fig5 begins to address this and provides some evidence that beta-catenin overexpression in a cell line modestly compensates for loss of CREPT and is complexed to CREPT in vivo (Fig 7A). Other evidence such as reduced nuclear beta-catenin in a colorectal cancer cell line after CREPT mutation (Fig 7B) is interesting but what is the effect of CREPT on proliferation in these cells, and might the changes observed be secondary ones?

Minor point: It is hard to see nuclear beta-catenin in the images in Fig7C-F. There seem few of these. The number of positive nuclei per crypt (Fig 7G) gives no indication if crypts are larger in the WT mice. If expressed as a frequency of positive nuclei would there still be significance?

The Discussion makes the claim that: "our results demonstrated that CREPT is required for Wnt signaling activation in tumor cells..." . Do the authors refer to SFig5 on HEK293Ts? Otherwise this cannot relate to any data in this manuscript and would seem without foundation.

Reviewer #2:

Remarks to the Author:

Yang and colleagues provide evidence for a role of CREPT in intestinal stem cell maintenance in the mouse. The authors show that deletion of CREPT results in loss of crypt basal stem cells and an inability to recover from irradiation - induced gut damage. Using transcriptome analysis, the authors identify perturbed Wnt signaling as a possible mediator of these effect and using reporter assays they demonstrate that CREPT is required for the activation of Wnt signaling targets. Based on immunostaining and western blot data, the authors propose that CREPT is required to maintain nuclear localization of beta catenin in the crypt stem cells.

The data is overall of good quality, clearly establishing a critical role for CREPT in the intestinal crypt. The assertion that CREPT is acting via modulation of Wnt signaling requires additional support, however. It is mostly derived from transcriptome profiling studies that indicate changes in Wnt target genes, and it would be important to support these observations with genetic studies that clearly demonstrate that the loss of Lgr5+ cells in CREPT mutants is a consequence of these transcriptional changes and not a consequence of simply apoptosis due to an unspecific role for CREPT in cell survival. The authors acknowledge in their discussion that they can't exclude a role for CREPT in DNA repair, and their data that GSK3 inhibition cannot fully rescue the loss of CREPT argues against a role for CREPT in maintaining b catenin nuclear localization (a strong up regulation in beta catenin levels should overcome the lack of CREPT if this was its sole function).

Further characterization and compelling data on the function of CREPT in Wnt signaling would be required to support publication.

Reviewer #3:

Remarks to the Author:

CREPT is required for Lgr5+ stem cells maintenance and controls intestinal regeneration

This report by Yang et al., describes the role of CREPT, an RNA pol II binding protein, in intestinal homeostasis and regeneration. From their experiments, the authors conclude that CREPT is necessary for the activation of Wnt signaling in the intestinal crypt and this instructive role of CREPT is necessary to maintain intestinal stem cell proliferation and tissue regeneration following damage. CREPT induces Wnt signaling activation by binding to B-Catenin and facilitating its nuclear localisation/activity.

CREPT function, including its binding to B-Catenin, has been previously characterized by the authors in the context of tumours but this is the first report on the role of CREPT in intestinal homeostasis and regeneration.

Major comments:

1- Vil-CREPT KO mice die at high ratios and the surviving ones have reduced size/weight. Given that Villin Cre is expressed during the earlier stages of intestinal development, these results may reflect a role of CREPT in intestinal development rather than adult tissue homeostasis.

Consistently, the use of a more restricted and inducible LGR5-Cre gives a much milder phenotype. However, the authors have not explored/commented on this:

a- Why are surviving animals smaller/leaner? Is intestinal growth and physiology (e.g. nutrient absorption) impaired in the Vil-CREPT KO mice?

b- What is the outcome of deleting CREPT using an inducible Vil-CRE (e.g. Villin-CRE-ER).

2- How do the authors interpret the increased Crypt thickness in the large intestine of Vil-CREPT KO mice (Figure S1E)?

3- The decrease in crypt cell proliferation and crypt/villi cell migration in the homeostatic small intestine of Vil-CREPT KO mice is not evident (Figure 1).

4- Clearly, CREPT is expressed beyond LGR5 +ve cells. How do the authors explain the differences in the phenotype of the Villin versus LGR5 CREPT knock out mice? Please see also related point 1.

5- Similarly, the intestinal phenotype of LGR5-CREPT knock out mice is weaker than that of LGR5

CREPT knock out organoids. How do the authors explain that?

6- Vil-CREPT loss leads to a very strong prevention of intestinal regeneration. What happens in the LGR5-CREPT knock out mice?

7- Is CREPT induced upon regeneration?

8- Authors mention that the role of CREPT is more prominent in regeneration than in homeostasis. Is that the case? Or is it due to the more acute need for proliferation following damage that makes the role of CREPT more evident?

9- In Figure 4H, the authors claim that LGR5-CREPTKO organoids have less EDU. This is not reflected in the data presented. I see equal amounts of EDU in the present crypts. The knockout organoids seem just smaller/have less crypts.

10- Quantifications should be provided for Figures S4H, I.

11- CREPT binds to B-Catenin in vivo and in vitro and is required to induce nuclear accumulation of B-Catenin:

a- The nuclear localization of B-Catenin is not evident from the data presented in Figure 7C-F. IHC with published protocols optimised for the detection of nuclear B-Catenin should be done.

b- Figure S5F: Exogenous CREPT/B-Catenin binding. There is a problem with this Western Blots. From top to bottom: Middle lane of the second blot shows a Myc-CREPT band following IP with anti-Myc antibody. However, Myc-CREPT is clearly not present in this condition as indicated in the labels at the top of the Western blot images and confirmed by the analysis of whole lysates (middle lane of the fourth blot).

Discussion: Lines 318-321. The authors point to the fact that expression of Cyclin D1 Myc, unlike other Wnt signaling targets, is not dependent on CREPT. Therefore, they conclude that those genes may not be required for intestinal regeneration. Myc is indeed essential for intestinal proliferation (Muncan et al., Mol. Cell. Biol. 2006; Cordero et al EMBO J 2012). Alternatively, Myc and CyclinD1 could be targets of other signaling pathways and clearly, their expression is not sufficient to compensate for the loss of CREPT and other Wnt targets in the intestine. The authors should revise this section of the discussion to consider other alternatives.

Overall, the text should be edited to correct typos and spelling mistakes.

Reviewer #4:

Remarks to the Author:

CREPT/RPRD1B, discovered in 2011 as a scaffold protein of human RNA polymerase II (refs. 27, 28) has been identified later as a proto-oncogene highly expressed in tumors. Studies in the Chang lab have revealed that CREPT, through its CTD-interacting domain, binds to both beta-catenin and TCF4, resulting in enhanced transcription of Wnt target genes, in particular Cyclin D1 and c-Myc, leading to upregulation of cell proliferation and invasion. In addition CREPT also binds to Aurora kinase B and regulates Cyclin B1 expression (ref. 24).

In the present study the Chang group uses mouse models to study the physiological consequences of deleting CREPT in villin-expressing intestinal epithelial cells (Vil-CREPTKO mice) or of conditionally (i.e. tamoxifen-induced) deletion of CREPT in Lgr5+ expressing intestinal stem cells (ISCs), both in living mice (Lgr5-CREPTKO mice) and in stem cell-derived intestinal organoids. The deletion of CREPT in mouse intestine caused many physiological changes, including reduction in size and body weight, retarded migration of enterocytes to the villus tip, a failure to regenerate after X-ray irradiation and DSS treatment, reduced Lgr5+ cell numbers at homeostasis, the partial loss of differentiated cell types (goblet cells, Paneth cells) and a decreased expression of Wnt target genes. It is concluded that CREPT is a novel regulator of ISCs and that CREPT deletion down-regulates Wnt signaling in ISCs by impaired beta-catenin accumulation in the nucleus during epithelial regeneration. However other/additional mechanisms of action, e.g. the promotion of DNA repair in ISCs after irradiation, are not excluded but not yet experimentally tested.

The study is technically sound and addresses an important issue, i.e. whether CREPT plays a physiological role in maintaining intestinal cell renewal and its capacity to regenerate after

chemical or radiation damage. Though not discussed, the outcome of the study, if extrapolable to human intestine, implies that downregulation of CREPT by microRNAs or other potential tumor suppressor approaches might cause serious adverse effects in the intestine.

Specific comments (major):

1. So far the role of CREPT in tumorigenesis has been studied in human tumors and tumor cell lines, whereas in the present study its physiological role is studied in mouse intestine and mouse organoids, but not in human tissues/cells (with the exception of DLD1 colorectal tumor cells; Fig. 7B). Ideally, one would like to see confirmation of some of the key findings of the mouse study in human intestine e.g. by knockdown of CREPT expression at the level of human intestinal organoids. If such data cannot be provided, the possibility of species differences in CREPT expression and mechanism of action between mouse and human intestine should at least be mentioned and discussed in the Introduction (e.g. l. 74) or Discussion section.
2. L. 96 and Fig. 1B: Are the crypts isolated from small or large intestine? What does "Intestines" mean: whole pieces of intestine, crypt+villous epithelial cells (e.g. mucosal scrapings), or villous cells only?
3. Fig. 1C, D: the smaller size and loss of body weight in the Vil-CREPTKO mice suggest that the digestion and uptake of nutrients in the intestinal villi is strongly reduced. On the other hand, the cell kinetics are changed but the villus number and structure in the jejunum were normal (l. 100 and Suppl. Fig. 1E). To explain this discrepancy, is there evidence for a reduced expression of villus markers (e.g. sucrase-isomaltase; SGLT1; lactase) in the CREPT-KO intestine?
4. L. 105, Suppl. Fig. 1E: If Ki67 is a reliable marker of proliferating cells, how do the authors explain that the number of Ki67-positive cells in the SI crypts was not different between WT and Vil-CREPTKO mice, whereas CREPT is shown later (l. 153) to be essential for the maintenance of Lgr5+ stem cells and Edu labeling of crypt cells is reduced in Vil-CREPTKO mice (Fig. 1G)?
5. L. 118, Fig. 1J: the claim that the number of lysozyme-stained Paneth cells is significantly decreased in Vil-CREPTKO mice, if based solely on the "representative" Fig. 1J, is not convincing. Visual inspection suggests that (possibly) the distribution of these cells, but not the number is changed. Is there a drop in the transcript levels of lysozyme or other Paneth cell markers in the intestine of Vil-CREPTKO mice, comparable to the downregulation of Paneth cell markers (Nox 1, Atoh1) reported later (l. 191) in CREPT-deleted Lgr5+ cells?
6. Fig. 2A: If Bmi1 is robustly expressed not only in +4 reserve stem cells but also in Lgr5+ cells (ref. 16), how do the authors explain that the expression of this ISC marker remains unaltered in Vil-CREPTKO mouse intestine whereas Lgr5+ expression (Fig. 2A) and the number of Lgr5+ expressing ISCs (Fig. 2C) is strongly reduced?
7. L. 198, Fig. 5A: Expression of the ISC signature genes was barely affected in the (remaining) Lgr5+-CREPTKO cells. However, as mentioned in l. 246, these stem cell signature genes are mainly regulated by Wnt signaling, which also controls the expression of cell cycle promoting genes and differentiation determining genes. How then can we envision that the latter genes are down-regulated in Lgr5+-CREPTKO cells, whereas the Wnt-controlled stem cell signature genes are not?
8. L. 310-321: The surprising finding in this study that Wnt targeting is impaired in regenerating intestine of Vil-CREPTKO mice but does not result in down-regulation of the Wnt target genes CCND1 (Cyclin D1) and c-Myc is interpreted by the authors as a sign of different Wnt signaling regulation between tumors and ISCs (l. 321). It would be of prime importance to know whether this represents a difference between species (mouse vs. human) or between tumors and non-malignant epithelium. In the first case, it would indicate that the mouse model is not an appropriate surrogate for (regenerating) human intestinal epithelium. In any case, because many other Wnt target genes are down-regulated in the mouse model, the outcome suggests that CCND1 and MYC do not function as Wnt target genes in mouse intestine. Is there any, more direct evidence supporting this observation?
9. Fig. 7G: The quantity of nuclear beta-catenin in the crypts of non-irradiated Vil-CREPTKO mice was not different from WT mice. On the other hand, the expression of cell cycle promoting genes in Lgr5-CREPTKO cells was strongly decreased (Fig. 4G), presumably by a defect in Wnt-beta catenin signaling. Is there any indication that the level of nuclear beta-catenin differs between Lgr5+-CREPTKO cells and WT Lgr5+ cells? If not, this would strengthen the notion that the

mechanism of Wnt-beta catenin signaling under physiological conditions differs importantly from Wnt-beta catenin signaling in irradiated/regenerating intestine.

Specific comments (minor):

1. Style and English in the text and figure legends is weak and needs multiple corrections (e.g. CREPT-deficiency mice=CREPT-deficient mice; intestine crypts = intestinal crypts etc.).
2. L. 81: retention in...Please complete the sentence.
3. L. 627: (F and G) should be replaced by(G and H); (H and I) by (I and J).
4. Legends, Suppl. Figs. 4 and 5: Presumably 5 and 2 mM CHIR99021 should be changed to 5 and 2 μ M?

Point to point response

Reviewer #1 (Remarks to the Author):

The manuscript by Chang and colleagues explores the role of a candidate cancer driver gene, CREPT, in maintaining intestinal stem (Lgr5+) cells (ISCs) and in controlling intestinal regeneration. Using intestinal specific deletion of CREPT the authors describe the resultant phenotypes, finding modest effects on the steady state and rather more dramatic effects when the gut is forced to into a regenerative phase by high dose irradiation. The initial motivation seems to have been to document the physiological role of CREPT in a whole tissue/animal setting because this has not yet been done. The authors find that stem cells as defined by a number of markers but mainly Lgr5 are underrepresented in adult intestinal epithelium following deletion of CREPT in the developing gut due to Cre-mediated deletion (VilCre model).

It is worth noting that CREPT is viewed as a cell cycle regulator that regulates directly (promotes binding of RNA pol II) or enhances transcription of a number of cell cycle regulatory genes. In this respect it is somewhat surprising that there is not more attention paid to the cell cycle related changes occurring in the intestinal epithelium. Is replication stalled; if so at what stage and are damage markers induced?

The authors present a significant amount of data that in the main seems to have performed to a good standard albeit with some missing and important technical details. Their conclusion about ISCs being maintained by CREPT is correct but how and when that maintenance is achieved remains somewhat elusive. The phenotypic characterization of steady and regenerative states seems good. Mechanistic insights are relatively superficial: molecular profiling tools are used almost exclusively and it is often unclear where whether they are describing the phenotype or elucidating it.

In the steady state CREPT KO mice show grossly normal small intestine but enlarged crypt epithelium in the colon. In both tissues proliferation is compromised and secretory and ISC are underrepresented. In an apparent adaptation cellular migration of cells along the crypt to villus axis is slowed. Of note the interpretation of reduced ISC numbers depends on cell counts detecting GFP expressed from the Lgr5-EGFP-iresCreERT2 mouse line made by Barker but with CREPT still deleted due to VilCre. The Lgr5 mice are notoriously mosaic in adults with some crypts expressing the transgene and others not. Given that the Villin is expressed during gut development it is possible that silenced patches are preferentially advantaged following deletion of CREPT. Some reassurance is needed in this regard before fully accepting the data shown in Fig2.

Minor point: the IF for CREPT shown in Fig1A or Suppl Fig1 C does not convince that there it is nuclear localized and seems to show strong apical staining throughout the crypt. These images contrast with Fig2D where nuclear staining is obviously present.

The sensitivity of the CREPT deficient mice to irradiation compared to WT seems clear. One point of concern is the wording describing how survival analyses were done (page 7) where mice are said to have died due to the irradiation. From an animal ethics standpoint mice receiving such treatments should be on enhanced monitoring schemes and culled when reaching a predefined threshold of departure from health. Reassurance on this point is required.

Minor point: The time post IR for the counts shown in Fig3 D and E should be stated.

The data in Fig4 and 5A describes transcriptional profiling of purified ISCs isolated following deletion of CREPT using the Lgr5CreERT2 model. Missing from the experimental description is the time post tamoxifen treatment when tissues were harvested (the mice receive 1mg a day for 5

days) and if the controls also received tamoxifen. This analysis is welcome and informative but some phenotypic analysis over time would have enriched it. (Do the number of Lgr5+ cells decrease with time following CREPT deletion?). The analysis shows minimal impact on most ISC associated regulatory genes but significant changes in transcriptional abundance of cell cycle associated genes.

The analysis of Gene Ontology terms lacks detail: were the 5 GO terms for downregulated genes and 4 for upregulated gene the top 5 and 4 that came from the analysis? If not the full ordering with their significance should be shown perhaps in supplementals. There are a lot of changes in cell cycle associated genes. Is it not surprising that these did not generate a significant grouping in the GO analysis?

The data and related text describing Fig5 is rather odd. This is an analysis of similar markers to those shown in Fig4 but now looking at whole crypt epithelium. There is considerable duplication with elements of Fig 1 and 2 in demonstrating that ISCs and secretory cells are under represented in the CREPT VilCre knockouts. The authors seem to accept this but to also refer to CREPT “suppressing expression” of these ISC genes. Yet Fig4 seems to make clear that these genes are largely not so regulated. The reduction in transcriptional abundance in the whole crypt analysis would seem merely to reflect the relative abundance of the cells present.

A striking feature of comparing Fig4G and Fig5E is that changes in cell cycle genes are near identical in both ISC and whole crypts. It is difficult not to feel that these effects are the predominant consequences of CREPT deletion, more so than the effects on ISCs.

The final sections of the manuscript search for mechanistic insight by probing the differential changes in transcriptional abundance in WT and CREPT deficient gut following irradiation. They identify that five trophic pathways (by GO term analysis) are implicated in the regenerative response

in both genotypes. (Again are these the top 5 terms obtained from the analysis or were they selected?) Operating on the hypothesis that CREPT deficiency might impede activation of one or more of these pathways the authors look for a confused response where genes that are upregulated in WT are downregulated in the knockout. They identify genes associated with Wnt signaling as showing this behavior. However it is not entirely clear what feature of the gene set enrichment analysis allows this selection as PI3K-AKT, Hippo and MAPK similarly have sets of genes behaving this way. Is the normalized enrichment score for Wnt really different from these others or are they all much the same? Are

negative regulators behaving differently from positive regulators?

Again one wonders the extent to which the analyses shown in Figure 6 is describing the phenotype rather than identifying the mediators of it. With that in mind some definitive and/or orthogonal validation is important. Figure 7/Suppl Fig5 begins to address this and provides some evidence that beta-catenin overexpression in a cell line modestly compensates for loss of CREPT and is complexed to CREPT in vivo (Fig 7A). Other evidence such as reduced nuclear beta-catenin in a colorectal cancer cell line after CREPT mutation (Fig 7B) is interesting but what is the effect of CREPT on proliferation in these cells, and might the changes observed be secondary ones?

Minor point: It is hard to see nuclear beta-catenin in the images in Fig7C-F. There seem few of these. The number of positive nuclei per crypt (Fig 7G) gives no indication if crypts are larger in the WT mice. If expressed as a frequency of positive nuclei would there still be significance?

The Discussion makes the claim that: “our results demonstrated that CREPT is required for Wnt signaling activation in tumor cells....” . Do the authors refer to SFig5 on HEK293Ts? Otherwise this cannot relate to any data in this manuscript and would seem without foundation.

Responses to Referee #1 (Author's Response)

Question: It is worth noting that CREPT is viewed as a cell cycle regulator that regulates directly (promotes binding of RNA pol II) or enhances transcription of a number of cell cycle regulatory genes. In this respect it is somewhat surprising that there is not more attention paid to the cell cycle related changes occurring in the intestinal epithelium. Is replication stalled; if so at what stage and are damage markers induced?

Response: We thank the reviewer for the advice of detecting the cell cycle. For this purpose, we isolated the crypt cells from the intestines of wild type and CREPT KO mice, because the proliferating cells majorly reside in crypts. We performed a FACS analysis to demonstrate cells in different cell cycles by using cell cycle dye propidium iodide to mark DNA. The results showed that CREPT deletion caused decreased cell numbers in S/G2/M phases. We observed 11.3% and 9.45% (WT) vs 6.52% and 6.89% (CREPT KO) S/G2/M phase cells. We now presented the results in Supplementary Fig. 1g. We also cited this figure in our revised text.

To determine if CREPT participates in the DNA damage repairing in the crypt cells, we detected several damage markers including cleaved Caspase 3 and γ H2A.X in non-irradiated and irradiated WT and Vil-CREPT^{KO} intestines. The results showed that the deletion of CREPT led certain alteration of these two markers at the protein level. Please refer to the result as attached here. We did not observe any DNA replication stalling but the DNA replication rate appeared slower after CREPT was deleted. This is another project in our lab. We are sorry that we could not be able to provide these results in this single paper. We thank the reviewer for his/her question, which leads to our direction to reveal the role of CREPT in DNA replication.

Figure for the reviewer. The cleaved caspase 3 and γ H2A.X expression in untreated (a) and irradiated (b) WT and Vil-CREPT^{KO} intestines.

Question: The authors present a significant amount of data that in the main seems to have performed to a good standard albeit with some missing and important technical details. Their conclusion about ISCs being maintained by CREPT is correct but how and when that maintenance is achieved remains somewhat elusive. The phenotypic characterization of steady and regenerative states seems good. Mechanistic insights are relatively superficial: molecular profiling tools are used almost exclusively and it is often unclear where whether they are describing the phenotype or elucidating it.

Response: We appreciate the reviewer for her/his comments on our data. For the missing technical details, we had some problems with space in the text. Now, we decided to enumerate the important points of experiment processes in the supplemental materials and methods, such as animal ethics standpoint and the experimental design of Tamoxifen treatment. We added the mice number of each experiment in figure legends.

For the functioning time of CREPT on ISCs, we speculate that CREPT maintains ISCs at the stages of embryo and adult. In our models, CREPT was deleted by Villin-Cre during gut development, and in adult by Lgr5-Cre. In both cases, the Lgr5⁺ ISCs were dramatically decreased.

For the molecular mechanisms on how ISCs were maintained by CREPT, we proposed that CREPT regulates Wnt signaling, in particular, to retain β -catenin in the nucleus. We employed RNA-seq analysis in our previous version of the manuscript (see Figure 4 and 5). Now we performed a CHIP-

seq experiment to confirm the regulation of Wnt targeted genes by CREPT. We now presented the results in Fig.6e and 6f. The results showed that CREPT occupied at the promoter and termination region (before the poly A site) of Wnt targeted genes including *Cd44* and *EphB1*. This occupancy pattern is similar to that of *CCND1* in tumor cells (see our Cancer Cell paper in 2012). All these results suggest that CREPT regulates Wnt targeted genes at the transcription level. We also revised our descriptions to clearly describe the results. We also elucidate our results in a separate statement manner.

Question: Of note the interpretation of reduced ISCs numbers depends on cell counts detecting GFP expressed from the Lgr5-EGFP-iresCreERT2 mouse line made by Barker but with CREPT still deleted due to VillCre. The Lgr5 mice are notoriously mosaic in adults with some crypts expressing the transgene and others not. Given that the Villin is expressed during gut development it is possible that silenced patches are preferentially advantaged following deletion of CREPT. Some reassurance is needed in this regard before fully accepting the data shown in Fig2.

Response: Indeed, the Lgr5 mice are mosaic with some Lgr5-GFP⁻ cells expressing the Lgr5 protein. The loss of Lgr5-GFP⁺ cells might be caused by the Villin-cre induced CREPT deletion during development. To circumvent this concern, we previously demonstrated the decreased numbers of Lgr5-GFP⁺ cells in adult mice with CREPT deletion driven by TAM-induced CRE activation. This could avoid the preferential advantage of silenced patches due to CREPT deletion during gut development.

To reassure the impact of CREPT on ISCs, we determined to examine the Lgr5-GFP⁺ cells in adult mice for 30 days by TAM treatment. In this model, we eliminated the possibility of Villin-cre preference. The results showed that the number of Lgr5-GFP⁺ cells at day 30 dropped to a quarter of that at day 0 post TAM treatment, correlated to the depletion of CREPT. We now presented the data in Fig. 2g-2i. We also cited these results in our revised text. All these data suggested CREPT regulated the proliferation of Lgr5⁺ cells, and the decrease of Lgr5⁺ cells in Vill-CREPT^{KO} mice was not caused by Lgr5-GFP⁻ silenced patches following CREPT deletion during development.

Question: Minor point: the IF for CREPT shown in Fig1A or Suppl Fig1 C does not convince that there it is nuclear localized and seems to show strong apical staining throughout the crypt. These images contrast with Fig2D where nuclear staining is obviously present.

Response: We optimized the staining protocol and re-stained the CREPT as showed previously in Fig1A or Suppl Fig1 C. We now presented the clear IF results in revised Fig. 1a and Supplementary Fig. 1c.

Question: The sensitivity of the CREPT deficient mice to irradiation compared to WT seems clear. One point of concern is the wording describing how survival analyses were done (page 7) where mice are said to have died due to the irradiation. From an animal ethics standpoint mice receiving such treatments should be on enhanced monitoring schemes and culled when reaching a predefined threshold of departure from health. Reassurance on this point is required.

Response: We appreciated the reviewer for concerning the animal ethics in our irradiation process. Our animal experiments were performed under the guidance of AAALAC (Association for Assessment and Accreditation of Laboratory Animal Care International) and the IACUC (Institutional Animal Care and Use Committee) of Tsinghua University. We observed the behavior and health condition of irradiated/DSS-treat mice every day. The mice were judged to be of departure from health and euthanized by CO₂, when the euthanasia thresholds meet the predefined criteria. We now revised the words and added the information in the supplemental materials and methods.

The animal ethics guidance was recapitulated in the revised manuscript.

Question: Minor point: time post IR for the counts shown in Fig3 D and E should be stated.

Response: Fig3 D and E showed the counts for reformed crypts 5 days post IR. We now labeled the time in the figures and also stated the time point in the legends.

Question: The data in Fig4 and 5A describes transcriptional profiling of purified ISCs isolated following deletion of CREPT using the Lgr5CreERT2 model. Missing from the experimental description is the time post tamoxifen treatment when tissues were harvested (the mice receive 1mg a day for 5 days) and if the controls also received tamoxifen. This analysis is welcome and informative but some phenotypic analysis over time would have enriched it. (Do the number of Lgr5⁺ cells decrease with time following CREPT deletion?). The analysis shows minimal impact on most ISC associated regulatory genes but significant changes in transcriptional abundance of cell cycle associated genes.

Response: We apologize for missing the experimental description of tamoxifen treatment. The Lgr5CreERT2 (as WT mice) group and Lgr5-GFP; CREPT^{fl/fl} mice group both received tamoxifen at 1mg a day for 5 consecutive days, and analyzed at different times. This experimental design was described in detail in the section of Materials and Methods. We agree that we should figure out if the number of Lgr5⁺ cells decreases with time following CREPT deletion. As stated before, we analyzed the number of Lgr5⁺ cells at day 1, 7, 14 and 30 post TAM treatment in Lgr5-GFP; CREPT^{fl/fl} mice. We now added the results from different time points in Fig. 2g-2i. We also revised the text by citing these new results. It was clear that the number of Lgr5⁺ cells decreases with time post tamoxifen treatment. The RNA-seq analysis showed minimal effect on the ISC associated regulatory genes. We speculated that this might be due to the remaining CREPT KO Lgr5⁺ ISCs which compensated the stem cell function under the situation of ISC loss. However, these remaining Lgr5⁺ ISCs have limited ability of proliferation.

Question: The analysis of Gene Ontology terms lacks detail: were the 5 GO terms for downregulated genes and 4 for upregulated gene the top 5 and 4 that came from the analysis? If not the full ordering with their significance should be shown perhaps in supplementals. There are a lot of changes in cell cycle associated genes. Is it not surprising that these did not generate a significant grouping in the GO analysis?

Response: We understand the concern about the Gene Ontology analyses. The GO analyses always gave us repeat and unrelated terms, such as

“endocrine pancreas development”, “pancreatic A cell differentiation” and “forebrain development” derived from our data. However, we actually analyzed the transcriptome of WT and CREPT KO Lgr5⁺ intestinal stem cells. This is surprising since the samples of intestines should not be related to those organs. We looked into the genes detail of these terms, and found all the genes in “endocrine pancreas development”, “pancreatic A cell differentiation” and “forebrain development” terms were included in “positive regulation of cell differentiation” term, which we showed in Fig. 4C. Similar to the analysis in Fig. 4C, the distracted terms, such as “sensory perception of sound” and “morphogenesis of embryonic epithelium”, from the GO analysis of up-regulated genes were excluded. This is due to the limitation of GO analysis. We ignored these unrelated terms. Now, the top 5 and 4 terms without distraction were shown in Fig. 4C and 4D, respectively. We also added the full ordering with their significance in according Source Data file.

CREPT is identified as a cell cycle related gene, and promotes the expression of cyclin D1, cyclin E and other cell cycle promoting genes in tumor cells. We speculate that CREPT is a key regulator of the proliferation of intestinal stem cells. However, the P value of cell cycle related term generated by GO analysis is lower than 0.05 (no significance), which is contrary to our conjecture. In our results, we indeed observe the down-regulation of Cdk4, Cdk5, and Ccna1. We reasoned that only 125 genes are included in cell cycle term of KEGG and DAVID website (https://www.gsea-msigdb.org/gsea/msigdb/cards/KEGG_CELL_CYCLE), where many cell cycle genes had not been analyzed. Therefore, we re-analyzed our data by using a 312-cell-cycle-gene list from QIAGEN website, and the difference was displayed in Fig. 4g.

Question: The data and related text describing Fig5 is rather odd. This is an analysis of similar markers to those shown in Fig4 but now looking at whole crypt epithelium. There is considerable duplication with elements of Fig 1 and 2 in demonstrating that ISCs and secretory cells are under represented in the CREPT VilCre knockouts. The authors seem to accept this but to also refer to CREPT “suppressing expression” of these ISC genes. Yet Fig4 seems to make clear that these genes are largely not so regulated. The

reduction in transcriptional abundance in the whole crypt analysis would seem merely to reflect the relative abundance of the cells present.

A striking feature of comparing Fig4G and Fig5E is that changes in cell cycle genes are near identical in both ISC and whole crypts. It is difficult not to feel that these effects are the predominant consequences of CREPT deletion, more so than the effects on ISCs.

Response: We thank the reviewer for correcting the confusion in our manuscript. Previous Fig.4 and Fig.5 were the results of gene profile analysis under different knockout conditions. The previous Fig.4 showed the analysis of sorted WT and CREPT KO Lgr5+ cells, and the previous Fig.5 analyzed the transcription profiles of WT and CREPT deleted crypt cells. To make the manuscript logical and clear, we now remove the previous Fig.5 to supplementary data as new supplementary Fig. 4, and deleted the repeated cell cycle analysis.

Question: The final sections of the manuscript search for mechanistic insight by probing the differential changes in transcriptional abundance in WT and CREPT deficient gut following irradiation. They identify that five trophic pathways (by GO term analysis) are implicated in the regenerative response in both genotypes. (Again are these the top 5 terms obtained from the analysis or were they selected?) Operating on the hypothesis that CREPT deficiency might impede activation of one or more of these pathways the authors look for a confused response where genes that are upregulated in WT are downregulated in the knockout. They identify genes associated with Wnt signaling as showing this behavior. However it is not entirely clear what feature of the gene set enrichment analysis allows this selection as PI3K-AKT, Hippo and MAPK similarly have sets of gene behaving this way. Is the normalized enrichment score for Wnt really different from these others or are they all much the same? Are negative regulators behaving differently from positive regulators?

Response: We apologize for the confusion. As stated before, the GO and KEGG analyses gave us repeat and unrelated terms here, such as “Drug metabolism”, “Pancreatic secretion” and “Vascular smooth muscle contraction”. Thus, we selected the top five related KEGG pathway terms from over 200 enriched terms (in revised supplementary Fig. 5b). For the

GSEA, the analysis software automatically considered these five pathways were majorly enriched in up-regulated genes after CREPT deletion. It is clear that the ES value of Wnt signal related genes was 0.23, while that of other was 0.28 to 0.37. The lowest value of ES in Wnt signal related genes indicated the most downregulated genes among these five pathways. The normalized enrichment scores (NES) showed similar trend for orders of different pathways. Therefore, we focused on Wnt signal pathway regulated by CREPT during intestinal regeneration. Also, these results showed that the negative regulators (enriched in the up regulated genes) were reversely correlated to the positive regulators (enriched in the down regulated genes) from supplementary Fig. 5c-5g.

Question: Again one wonders the extent to which the analyses shown in Figure 6 is describing the phenotype rather than identifying the mediators of it. With that in mind some definitive and/or orthogonal validation is important.

Response: We thank the reviewer for the advice in Figure 6 (revised Fig. 5). As mentioned before, we now added the CHIP-seq results to reveal the mediators of CREPT regulation in Wnt signaling pathway. The results showed that CREPT bound to the promotor and termination regions of several Wnt target genes.

Question: Other evidence such as reduced nuclear beta-catenin in a colorectal cancer cell line after CREPT mutation (Fig 7B) is interesting but what is the effect of CREPT on proliferation in these cells, and might the changes observed be secondary ones?

Response: Previous Fig. 7B is now presented as Fig. 6g. We have observed that depletion of CREPT impaired the nuclear β -catenin in DLD-1 cells. This reviewer was interested in the proliferation status of the cells used in the experiment demonstrated in Fig. 6g. Indeed, the deletion of CREPT led to decreased proliferation, as observed in other cells in our previous papers (Lu, cancer cell, 2012; Zhang, JBC, 2014). We examined the clone formation of the DLD-1 cells and found that the deletion of CREPT dramatically reduced the clone numbers. We now present the results for your

reference. We would like not to present these results in this manuscript as they represent a repeated phenotype.

Figure for reviewer. Clones formed by 1000, 500 and 200 WT and CREPT KO DLD1 cells as indicated.

To address whether the reduced β -catenin could be the second effect under CREPT deletion, we presented a direct interaction of CREPT and β -catenin (Fig. 6e-6f). Also, deletion of CREPT did not affect the expression of β -catenin (Fig. 6g, total lysate, Supplementary Fig. 6a). We further observed that overexpression of CREPT promoted the β -catenin accumulation in the nucleus (see Figure for reviewer, immunostaining). All the results suggest that the role of CREPT on β -catenin accumulation in the nucleus is due to the interaction of CREPT and β -catenin. We apologize that we could not be able to decipher the detailed interaction mechanism of CREPT and β -catenin in this manuscript. We appreciate the reviewer for this question to direct our project in the future.

Figure for reviewer. CREPT promoted the nuclear retention of β -catenin. Human non-malignant intestinal epithelial cell line NCM 460 cells were stimulated with Wnt3a, and aberrant CREPT expression led to obvious nuclear retention of β -catenin.

Question: It is hard to see nuclear beta-catenin in the images in Fig7C-F. There seem few of these. The number of positive nuclei per crypt (Fig 7G) gives no indication if crypts are larger in the WT mice. If expressed as a frequency of positive nuclei would there still be significance?

Response: We performed IHC staining on tissue slices of irradiated WT and Vil-CREPT^{KO} intestines, which gave us much clearer nuclear β -catenin staining images. Now, we presented these images in revised Fig. 6h-6k. We also changed our statistic strategy from the number of positive nuclei per crypt to the number of nuclear β -catenin positive crypts per 1mm intestine. The quantitative presentation is now shown in Fig. 6l. These results are clear to demonstrate nuclear β -catenin.

Question: The Discussion makes the claim that: “our results demonstrated that CREPT is required for Wnt signaling activation in tumor cells....”. Do the authors refer to SFig5 on HEK293Ts? Otherwise this cannot relate to any data in this manuscript and would seem without foundation.

Response: We apologize for the confusion. The sentence “our results demonstrated that CREPT is required for Wnt signaling activation in tumor cells” refers to the previous data published by our lab (Zhang, JBC, 2012). Now, we added this reference and revised our statement in the text.

--

Reviewer #2 (Remarks to the Author):

Yang and colleagues provide evidence for a role of CREPT in intestinal stem cell maintenance in the mouse. The authors show that deletion of CREPT results in loss of crypt basal stem cells and an inability to recover from irradiation - induced gut damage. Using transcriptome analysis, the authors identify perturbed Wnt signaling as a possible mediator of these effect and using reporter assays they demonstrate that CREPT is required for the activation of Wnt signaling targets. Based on immunostaining and western blot data, the authors propose that CREPT is required to maintain nuclear localization of beta catenin in the crypt stem cells.

The data is overall of good quality, clearly establishing a critical role for CREPT in the intestinal crypt. The assertion that CREPT is acting via modulation of Wnt signaling requires additional support, however. It is mostly derived from transcriptome profiling studies that indicate changes in Wnt target genes, and it would be important to support these observations with genetic studies that clearly demonstrate that the loss of Lgr5+ cells in CREPT mutants is a consequence of these transcriptional changes and not a consequence of simply apoptosis due to an unspecific role for CREPT in cell survival. The authors acknowledge in their discussion that they can't exclude a role for CREPT in DNA repair, and their data that GSK3 inhibition cannot fully rescue the loss of CREPT argues against a role for CREPT in maintaining b catenin nuclear localization (a strong up regulation in beta catenin levels should overcome the lack of CREPT if this was its sole function).

Further characterization and compelling data on the function of CREPT in Wnt signaling would be required to support publication.

Responses to Referee #2 (Author's Response):

Question: Yang and colleagues provide evidence for a role of CREPT in intestinal stem cell maintenance in the mouse. The authors show that deletion of CREPT results in loss of crypt basal stem cells and an inability to recover from irradiation - induced gut damage. Using transcriptome analysis, the authors identify perturbed Wnt signaling as a possible mediator of these effect and using reporter assays they demonstrate that CREPT is required for the activation of Wnt signaling targets. Based on immunostaining and western blot data, the authors propose that CREPT is required to maintain nuclear localization of beta catenin in the crypt stem cells.

The data is overall of good quality, clearly establishing a critical role for CREPT in the intestinal crypt. The assertion that CREPT is acting via modulation of Wnt signaling requires additional support, however. It is mostly derived from transcriptome profiling studies that indicate changes in Wnt target genes, and it would be important to support these observations

with genetic studies that clearly demonstrate that the loss of Lgr5⁺ cells in CREPT mutants is a consequence of these transcriptional changes and not a consequence of simply apoptosis due to an unspecific role for CREPT in cell survival. The authors acknowledge in their discussion that they can't exclude a role for CREPT in DNA repair, and their data that GSK3 inhibition cannot fully rescue the loss of CREPT argues against a role for CREPT in maintaining β catenin nuclear localization (a strong up regulation in β catenin levels should overcome the lack of CREPT if this was its sole function).

Further characterization and compelling data on the function of CREPT in Wnt signaling would be required to support publication.

Response: We thank the reviewer for his/her appreciation of our data quality. We agree with that we should the support transcriptome profiling results with genetic studies. We now performed a ChIP-seq experiment to confirm the genetic regulation of CREPT in Wnt signaling pathway. The result showed that CREPT occupied the promotor and termination region of Wnt target genes. We added the results in Figure 5e and 5f. We also cited the results in the revised text. In addition, we examined the apoptosis marker cleaved caspase 3 in the intestines of WT and CREPT KO mice, and no significant difference was observed. We now present the results for your reference. These results suggested that the loss of Lgr5⁺ cells in CREPT KO mice is a consequence of transcriptional changes and not a consequence of simply apoptosis. We also revised the manuscript text accordingly.

Figure for reviewer. The protein expression of cleaved caspase 3 WT and Vil-CREPT^{KO} intestines.

To address the reason why GSK3 β inhibition cannot fully rescue the loss of CREPT, we speculate that the concentration of CHIR99021 previously used was insufficient in our previous luciferase reporter experiment. We previously treated cells with 2 μ M CHIR99021 for 4 h, and the luciferase activity of inhibitor treated cells is less than 10 times higher compared to that of DMSO treated cells. We now treated the cells with 5 μ M CHIR99021 for 4 h, and the luciferase activity of inhibitor treated cells increased for more than 100 times compared to that of DMSO treated cells (see revised Fig.6d). In addition, the higher concentration of GSK3 β inhibitor now fully rescued the loss of CREPT (revised Fig.6d). We have updated the presentation in the revised manuscript. All the results suggest that CREPT activates Wnt signaling through β -catenin translocalization in the nucleus.

--

Reviewer #3 (Remarks to the Author):

CREPT is required for Lgr5+ stem cells maintenance and controls intestinal regeneration

This report by Yang et al., describes the role of CREPT, an RNA pol II binding protein, in intestinal homeostasis and regeneration. From their experiments, the authors conclude that CREPT is necessary for the activation of Wnt signaling in the intestinal crypt and this instructive role of CREPT is necessary to maintain intestinal stem cell proliferation and tissue regeneration following damage. CREPT induces Wnt signaling activation by binding to B-Catenin and facilitating its nuclear localisation/activity.

CREPT function, including its binding to B-Catenin, has been previously characterized by the authors in the context of tumours but this is the first report on the role of CREPT in intestinal homeostasis and regeneration.

Major comments:

1- Vil-CREPT KO mice die at high ratios and the surviving ones have reduced size/weight. Given that Villin Cre is expressed during the earlier stages of intestinal development, these results may reflect a role of CREPT in intestinal development rather than adult tissue homeostasis. Consistently,

the use of a more restricted and inducible LGR5-Cre gives a much milder phenotype. However, the authors have not explored/commented on this:

a- Why are surviving animals smaller/leaner? Is intestinal growth and physiology (e.g. nutrient absorption) impaired in the Vil-CREPT KO mice?

b- What is the outcome of deleting CREPT using an inducible Vil-CRE (e.g. Villin-CRE-ER).

2- How do the authors interpret the increased Crypt thickness in the large intestine of Vil-CREPT KO mice (Figure S1E)?

3- The decrease in crypt cell proliferation and crypt/villi cell migration in the homeostatic small intestine of Vil-CREPT KO mice is not evident (Figure 1).

4- Clearly, CREPT is expressed beyond LGR5 +ve cells. How do the authors explain the differences in the phenotype of the Villin versus LGR5 CREPT knock out mice? Please see also related point 1.

5- Similarly, the intestinal phenotype of LGR5-CREPT knock out mice is weaker than that of LGR5 CREPT knock out organoids. How do the authors explain that?

6- Vil-CREPT loss leads to a very strong prevention of intestinal regeneration. What happens in the LGR5-CREPT knock out mice?

7- Is CREPT induced upon regeneration?

8- Authors mention that the role of CREPT is more prominent in regeneration than in homeostasis. Is that the case? Or is it due to the more acute need for proliferation following damage that makes the role of CREPT more evident?

9- In Figure 4H, the authors claim that LGR5-CREPTKO organoids have less EDU. This is not reflected in the data presented. I see equal amounts of EDU in the present crypts. The knockout organoids seem just smaller/have less crypts.

10- Quantifications should be provided for Figures S4H, I.

11- CREPT binds to B-Catenin in vivo and in vitro and is required to induce nuclear accumulation of B-Catenin:

a- The nuclear localization of B-Catenin is not evident from the data presented in Figure 7C-F. IHC with published protocols optimised for the detection of nuclear B-Catenin should be done.

b- Figure S5F: Exogenous CREPT/B-Catenin binding. There is a problem with this Western Blots. From top to bottom: Middle lane of the second blot shows a Myc-CREPT band following IP with anti-Myc antibody. However, Myc-CREPT is clearly not present in this condition as indicated in the labels at the top of the Western blot images and confirmed by the analysis of whole lysates (middle lane of the fourth blot).

Discussion: Lines 318-321. The authors point to the fact that expression of Cyclin D1 Myc, unlike other Wnt signaling targets, is not dependent on CREPT. Therefore, they conclude that those genes may not be required for intestinal regeneration. Myc is indeed essential for intestinal proliferation (Muncan et al., Mol. Cell. Biol. 2006; Cordero et al EMBO J 2012). Alternatively, Myc and CyclinD1 could be targets of other signaling pathways and clearly, their expression is not sufficient to compensate for the loss of CREPT and other Wnt targets in the intestine. The authors should revise this section of the discussion to consider other alternatives.

Overall, the text should be edited to correct typos and spelling mistakes.

Responses to Referee #3 (Author's Response):

Question: 1- Vil-CREPT KO mice die at high ratios and the surviving ones have reduced size/weight. Given that Villin Cre is expressed during the earlier stages of intestinal development, these results may reflect a role of CREPT in intestinal development rather than adult tissue homeostasis. Consistently, the use of a more restricted and inducible LGR5-Cre gives a much milder phenotype. However, the authors have not explored/commented on this:

a- Why are surviving animals smaller/leaner? Is intestinal growth and physiology (e.g. nutrient absorption) impaired in the Vil-CREPT KO mice?

b- What is the outcome of deleting CREPT using an inducible Vil-CRE (e.g. Villin-CRE-ER).

Response: We agree with that CREPT may play an important role in intestinal development, since CREPT deletion mediated by Villin-cre led to a smaller size and a low birth rate. In adult mice, CREPT knockout intestinal epithelium behaved normally in homeostasis, but were sensitive to physical and chemical challenges (see revised Fig. 3 and supplementary Fig. 3). Thus, we postulate that CREPT functions as a regulator for intestinal epithelium during gut development and in adult mice.

For question a, we apologize for missing the important information. The smaller sizes of surviving Vil-CREPT^{KO} mice may be due to the reduced differentiation of secretory cells and decreased nutrient uptake genes on CREPT KO intestinal epithelium. IHC staining experiments showed that the number of Paneth cells (Fig. 1j) and alcian blue stained goblet cells (Fig. 1k, 1l and revised Supplementary Fig.1i) were significantly decreased in Vil-CREPT^{KO} mice. The RNA-seq data also showed down-regulated marker genes of enteroendocrine cells, goblet cells and Paneth cells. The expression of genes related to nutrient uptake, such as Slc5a1 (sodium/glucose cotransporter), Slc2a5 (fructose transporter), Sis (sucrase-isomaltase), and Mgam (maltase-glucoamylase), were down-regulated in the intestines of Vil-CREPT^{KO} mice. However, the expression of the marker genes of absorptive cells were barely affected by the CREPT loss. These data demonstrated that the smaller size of CREPT KO mice might be due to the reduced quantity of secretory cells and down-regulated expression of nutrient uptake genes on intestinal epithelium. However, we did not dig into the role of CREPT in nutrition metabolism in this paper due to the limited space but we added sentences for this possibility in the discussion section. We will figure out this function of CREPT during the gut development and nutrition absorption in the future.

	WT		KO	
	#1	#2	#1	#2
Slc5a1	1640	1547	657	683
Sis	1003	600	459	642
Mgam	668	680	318	347
Slc2a5	245	257	67	73

Figure for reviewer. The FPKM of Slc5a1, Slc2a5, Sis, and Mgam of WT and Vil-CREPT^{KO} intestines from RNA-seq data.

For question b, we do not have the Villin-CRE-ER mouse line. We apologize for that. To compensate this shortage, we performed DSS treatment experiment on cre-ERT2; CREPT^{fl/fl} mice, which were induced by tamoxifen at 2 mg a day for 5 consecutive days for CREPT deletion in whole mouse body. The WT and cre-ERT2-CREPT KO mice were treated with 2.5% DSS for 7 days, and harvested at the day 3 post the DSS treatment. The results showed that the CREPT KO mice lost more weights than WT mice, and the architecture of colorectal crypts was recovered in WT mice but remained destroyed in Vil-CREPTKO mice at day 3 after DSS withdrawal. These results were in consistence with that from Vil-CREPT^{KO} mice. This model excludes the effect of Vil-Cre CREPT KO during gut development. We now presented the data from cre-ERT2-CREPT KO mice for your reference here.

Figure for reviewer. (a) Schematic diagram showing the treatment in cre-ERT2 (WT) and cre-ERT2; CREPT^{fl/fl} (ERT2-CREPT KO) mice. (b) The loss of DSS treated WT and ERT2-CREPT KO mice body weights. n≥4 mice per genotype. (c) Representative H&E-stained sections of colons from WT and ERT2-CREPT KO mice at day 3 after DSS treatment. Scale bars, 100 μm.

Question: 2- How do the authors interpret the increased Crypt thickness in the large intestine of Vil-CREPT KO mice (Figure S1E)?

Response: The increased crypt thickness may be due to the alteration of the morphology. As the proliferating cells were decreased (decreased Ki67+ cells in the large intestine, in supplementary Fig. 1f) and the limited numbers

of cells became differentiated, the overall phenotype is the enlarged crypt thickness. However, we are unable to decipher the detailed mechanism as this manuscript is mainly focused on the role of CREPT in small intestines rather than the large intestines. The role of CREPT in large intestines will be our next project in the future.

Question: 3- The decrease in crypt cell proliferation and crypt/villi cell migration in the homeostatic small intestine of Vil-CREPT KO mice is not evident (Figure 1).

Response: To show the migration difference, we quantified the EdU labelled cells to demonstrate the decrease in crypt cell proliferation (Fig. 1g), and measured the migration distances of EdU labelled cells in different times (Fig. 1h and 1i). We believe these data would support the observation of reduced crypt cell proliferation and crypt/villi cell migration in the homeostatic small intestine of Vil-CREPT KO mice.

Question: 4- Clearly, CREPT is expressed beyond LGR5 +ve cells. How do the authors explain the differences in the phenotype of the Villin versus LGR5 CREPT knock out mice? Please see also related point 1.

Response: We postulate that CREPT is expressed beyond Lgr5+ cells, and knocking out CREPT has a greater impact on the intestinal epithelium. Villin-cre induced CREPT deletion before birth affects the gut homeostasis during development. On the other hand, the Lgr5 mice are mosaic in adults with some crypts expressing the transgene, which is experimentally demonstrated in our studies (Lgr5-GFP staining of WT mice in Fig. 2b). CREPT is deleted in a part of Lgr5+ cells of Lgr5-CREPT^{KO} mice while in all intestinal epithelial cells of Villin-CREPT^{KO} mice, leading to different experimental results. We reasoned the embryo expression of CREPT is critical for the difference of two mouse models. We now added this point in our discussion.

Question: 5- Similarly, the intestinal phenotype of LGR5-CREPT knock out mice is weaker than that of LGR5 CREPT knock out organoids. How do the authors explain that?

Response: This difference might be due to the mosaic feature of the LGR5-cre induced CREPT deletion and the homogenous organoids *in vitro*. In the *in vivo* condition, various mesenchymal cells and tissue connective cells can support the growth of CREPT knock-out stem cells. However, under the situation of organoid culture, since we isolated the epithelial cells, not the mixture of the intestines, no more mesenchymal cells and tissue connective cells were provided and the CREPT deletion stem cells failed to grow in the absence of supporting cells *in vitro*.

Question: 6- Vil-CREPT loss leads to a very strong prevention of intestinal regeneration. What happens in the LGR5-CREPT knock out mice?

Response: We treated the WT and Lgr5-CREPT^{KO} mice with 10 Gy X-Ray irradiation, and the results showed no much difference in the intestinal regeneration ability of WT and Lgr5-CREPT^{KO} mice. This is in consistence to the weaker phenotype of Lgr5-CREPT^{KO} mice (as we explained in Question 4 and 5). We reasoned two aspects as 1) deletion of CREPT by villin cre initiated in the embryo stage but the Lgr5 cre started in the adult; 2) the Lgr5+ cre frequently had the mosaic feature. We have discussed this question in our revised manuscript.

Question: 7- Is CREPT induced upon regeneration?

Response: The CREPT mRNA level of irradiated intestines was similar to that of untreated ones (see the result in Fig. 5d). It appeared that CREPT was not induced upon regeneration in the model.

Question: 8- Authors mention that the role of CREPT is more prominent in regeneration than in homeostasis. Is that the case? Or is it due to the more acute need for proliferation following damage that makes the role of CREPT more evident?

Response: We agree that the acute need for proliferation following damage may make the role of CREPT more evident. CREPT is required for the Wnt signaling activation. The expression of Wnt target genes was up-regulated during the intestinal regeneration process, but remained grounded state in homeostasis (revised Fig. 5c). Thus, the role of CREPT is more evident in intestinal regeneration. However, CREPT is not dispensable in intestinal epithelial cells during homeostasis. We demonstrated the decreased numbers of Lgr5-GFP⁺ cells in adult mice with CREPT deletion driven by TAM-induced CRE activation. To address this question, we now performed an experiment by examining the Lgr5⁺-GFP cells post tamoxifen induced CREPT deletion. We presented the results in Fig. 2g-2i. The results showed that the number of Lgr5-GFP⁺ cells at day 30 dropped to a quarter of that at day 1 post TAM treatment, correlated to the depletion of CREPT. These data demonstrated that CREPT maintained the quantity of Lgr5⁺ stem cells during homeostasis. We have revised the manuscript text accordingly.

Question: 9- In Figure 4H, the authors claim that LGR5-CREPTKO organoids have less EDU. This is not reflected in the data presented. I see equal amounts of EDU in the present crypts. The knockout organoids seem just smaller/have less crypts.

Response: We apologize for the confusion. We intended to point out the decreased number of crypts in CREPT deleted organoids compared to WT ones, since EdU labelled cells were restricted in crypts (in revised Fig. 4h). We agree that the amounts of EdU⁺ cells in WT and CREPT deleted crypts seemed equal. Now, we revised the describing words in the main text.

Question: 10- Quantifications should be provided for Figures S4H, I.

Response: We added the quantification for previous Figures S4H and S4I (revised supplemental Fig.5h and 5i) in revised supplemental Fig. 5j.

Question: 11- CREPT binds to B-Catenin in vivo and in vitro and is required to induce nuclear accumulation of B-Catenin:

a- The nuclear localization of B-Catenin is not evident from the data presented in Figure 7C-F. IHC with published protocols optimized for the detection of nuclear B-Catenin should be done.

b- Figure S5F: Exogenous CREPT/B-Catenin binding. There is a problem with this Western Blots. From top to bottom: Middle lane of the second blot shows a Myc-CREPT band following IP with anti-Myc antibody. However, Myc-CREPT is clearly not present in this condition as indicated in the labels at the top of the Western blot images and confirmed by the analysis of whole lysates (middle lane of the fourth blot).

Response: a-We now replaced the IF staining experiments of Fig. 6 with IHC staining ones on tissue slices of irradiated WT and Vil-CREPT^{KO} intestines, which gave us much clearer nuclear β -catenin staining images. Now, we presented these images in revised Fig. 6h-6k. We also changed our statistic strategy from the number of nuclear β -catenin positive nuclei per crypt to the number of nuclear β -catenin positive crypts per 1mm intestine, and more evident nuclear localization of β -catenin was shown in revised Fig. 6l.

b- Now the previous Figure S5F is presented as Fig. 6f. We apologize for the labelling mistakes. Now we confirmed and changed the labels of the Western blot images in revised Fig. 6f, and all the uncropped blots were included in supplementary Fig. 7.

Question: Discussion: Lines 318-321. The authors point to the fact that expression of Cyclin D1 Myc, unlike other Wnt signaling targets, is not dependent on CREPT. Therefore, they conclude that those genes may not be required for intestinal regeneration. Myc is indeed essential for intestinal proliferation (Muncan et al., Mol. Cell. Biol. 2006; Cordero et al EMBO J 2012). Alternatively, Myc and CyclinD1 could be targets of other signaling pathways and clearly, their expression is not sufficient to compensate for the loss of CREPT and other Wnt targets in the intestine. The authors should revise this section of the discussion to consider other alternatives.

Response: We thank the reviewer for the correction. In our study, GSEA analysis showed major up-regulations in PI3K-AKT, Hippo, MAPK and TGF β signaling pathways in Vil-CREPT^{KO} intestines (revised

Supplementary Fig.5d-5g). Cyclin D1 and c-Myc were the downstream target genes of these signaling pathways. It is possible that cyclin D1 and c-Myc were up-regulated through these signaling pathways rather than Wnt signaling. We have now added a main paragraph to discuss this question in our revised discussion section. We also added the citations of the references related to the Myc function.

In addition, we checked and revised the typos and spelling mistakes in the manuscript. Thanks to the reviewer again for pointing out these mistakes.

Reviewer #4 (Remarks to the Author):

CREPT/RPRD1B, discovered in 2011 as a scaffold protein of human RNA polymerase II (refs. 27, 28) has been identified later as a proto-oncogene highly expressed in tumors. Studies in the Chang lab have revealed that CREPT, through its CTD-interacting domain, binds to both beta-catenin and TCF4, resulting in enhanced transcription of Wnt target genes, in particular Cyclin D1 and c-Myc, leading to upregulation of cell proliferation and invasion. In addition CREPT also binds to Aurora kinase B and regulates Cyclin B1 expression (ref. 24).

In the present study the Chang group uses mouse models to study the physiological consequences of deleting CREPT in villin-expressing intestinal epithelial cells (Vil-CREPTKO mice) or of conditionally (i.e. tamoxifen-induced) deletion of CREPT in Lgr5⁺ expressing intestinal stem cells (ISCs), both in living mice (Lgr5-CREPTKO mice) and in stem cell-derived intestinal organoids. The deletion of CREPT in mouse intestine caused many physiological changes, including reduction in size and body weight, retarded migration of enterocytes to the villus tip, a failure to regenerate after X-ray irradiation and DSS treatment, reduced Lgr5⁺ cell numbers at homeostasis, the partial loss of differentiated cell types (goblet cells, Paneth cells) and a decreased expression of Wnt target genes. It is concluded that CREPT is a novel regulator of ISCs and that CREPT deletion down-regulates Wnt signaling in ISCs by impaired beta-catenin accumulation in the nucleus during epithelial regeneration.

However other/additional mechanisms of action, e.g. the promotion of DNA repair in ISCs after irradiation, are not excluded but not yet experimentally tested.

The study is technically sound and addresses an important issue, i.e. whether CREPT plays a physiological role in maintaining intestinal cell renewal and its capacity to regenerate after chemical or radiation damage. Though not discussed, the outcome of the study, if extrapolable to human intestine, implies that downregulation of CREPT by microRNAs or other potential tumor suppressor approaches might cause serious adverse effects in the intestine.

Specific comments (major):

1. So far the role of CREPT in tumorigenesis has been studied in human tumors and tumor cell lines, whereas in the present study its physiological role is studied in mouse intestine and mouse organoids, but not in human tissues/cells (with the exception of DLD1 colorectal tumor cells; Fig. 7B). Ideally, one would like to see confirmation of some of the key findings of the mouse study in human intestine e.g. by knockdown of CREPT expression at the level of human intestinal organoids. If such data cannot be provided, the possibility of species differences in CREPT expression and mechanism of action between mouse and human intestine should at least be mentioned and discussed in the Introduction (e.g. l. 74) or Discussion section.
2. L. 96 and Fig. 1B: Are the crypts isolated from small or large intestine? What does “Intestines” mean: whole pieces of intestine, crypt+villous epithelial cells (e.g. mucosal scrapings), or villous cells only?
3. Fig. 1C, D: the smaller size and loss of body weight in the Vil-CREPTKO mice suggest that the digestion and uptake of nutrients in the intestinal villi is strongly reduced. On the other hand, the cell kinetics are changed but the villus number and structure in the jejunum were normal (l. 100 and Suppl. Fig. 1E). To explain this discrepancy, is there evidence for a reduced expression of villus markers (e.g. sucrase-isomaltase; SGLT1; lactase) in the CREPT-KO intestine?
4. L. 105, Suppl. Fig. 1E: If Ki67 is a reliable marker of proliferating cells, how do the authors explain that the number of Ki67-positive cells in the SI

crypts was not different between WT and Vil-CREPTKO mice, whereas CREPT is shown later (l. 153) to be essential for the maintenance of Lgr5+ stem cells and Edu labeling of crypt cells is reduced in Vil-CREPTKO mice (Fig. 1G)?

5. L. 118, Fig. 1J: the claim that the number of lysozyme-stained Paneth cells is significantly decreased in Vil-CREPTKO mice, if based solely on the “representative” Fig. 1J, is not convincing. Visual inspection suggests that (possibly) the distribution of these cells, but not the number is changed. Is there a drop in the transcript levels of lysozyme or other Paneth cell markers in the intestine of Vil-CREPTKO mice, comparable to the downregulation of Paneth cell markers (Nox 1, Atoh1) reported later (l. 191) in CREPT-deleted Lgr5+ cells?

6. Fig. 2A: If *Bmi1* is robustly expressed not only in +4 reserve stem cells but also in Lgr5+ cells (ref. 16), how do the authors explain that the expression of this ISC marker remains unaltered in Vil-CREPTKO mouse intestine whereas Lgr5+ expression (Fig. 2A) and the number of Lgr5+ expressing ISCs (Fig. 2C) is strongly reduced?

7. L. 198, Fig. 5A: Expression of the ISC signature genes was barely affected in the (remaining) Lgr5+-CREPTKO cells. However, as mentioned in l. 246, these stem cell signature genes are mainly regulated by Wnt signaling, which also controls the expression of cell cycle promoting genes and differentiation determining genes. How then can we envision that the latter genes are down-regulated in Lgr5+-CREPTKO cells, whereas the Wnt-controlled stem cell signature genes are not?

8. L. 310-321: The surprising finding in this study that Wnt targeting is impaired in regenerating intestine of Vil-CREPTKO mice but does not result in down-regulation of the Wnt target genes CCND1 (Cyclin D1) and c-Myc is interpreted by the authors as a sign of different Wnt signaling regulation between tumors and ISCs (l. 321). It would be of prime importance to know whether this represents a difference between species (mouse vs. human) or between tumors and non-malignant epithelium. In the first case, it would indicate that the mouse model is not an appropriate surrogate for (regenerating) human intestinal epithelium.

In any case, because many other Wnt target genes are down-regulated in the mouse model, the outcome suggests that CCND1 and MYC do not function

as Wnt target genes in mouse intestine. Is there any, more direct evidence supporting this observation?

9. Fig. 7G: The quantity of nuclear beta-catenin in the crypts of non-irradiated Vil-CREPTKO mice was not different from WT mice. On the other hand, the expression of cell cycle promoting genes in Lgr5-CREPTKO cells was strongly decreased (Fig. 4G), presumably by a defect in Wnt-beta catenin signaling. Is there any indication that the level of nuclear beta-catenin differs between Lgr5⁺-CREPTKO cells and WT Lgr5⁺ cells? If not, this would strengthen the notion that the mechanism of Wnt-beta catenin signaling under physiological conditions differs importantly from Wnt-beta catenin signaling in irradiated/regenerating intestine.

Specific comments (minor):

1. Style and English in the text and figure legends is weak and needs multiple corrections (e.g. CREPT-deficiency mice=CREPT-deficient mice; intestine crypts = intestinal crypts etc.).
2. L. 81: retention in...Please complete the sentence.
3. L. 627: (F and G) should be replaced by(G and H); (H and I) by (I and J).
4. Legends, Suppl. Figs. 4 and 5: Presumably 5 and 2 mM CHIR99021 should be changed to 5 and 2 uM?

Responses to Referee #4 (Author's Response)

Question: 1. So far, the role of CREPT in tumorigenesis has been studied in human tumors and tumor cell lines, whereas in the present study its physiological role is studied in mouse intestine and mouse organoids, but not in human tissues/cells (with the exception of DLD1 colorectal tumor cells; Fig. 7B). Ideally, one would like to see confirmation of some of the key findings of the mouse study in human intestine e.g. by knockdown of CREPT expression at the level of human intestinal organoids. If such data cannot be provided, the possibility of species differences in CREPT expression and mechanism of action between mouse and human intestine should at least be mentioned and discussed in the Introduction (e.g. l. 74) or Discussion section.

Response: Thanks for the reviewer's excellent suggestion. We performed such an experiment by using human organoids. To deplete CREPT protein, we used a PROTAC, named PRTC, which was developed for specifically targeting CREPT and mediating its degradation. We have proved that PRTC worked in human tumor cells (Ma D, et al. 2020). We now employed PRTC to treat human colorectal organoids (see revised Supplementary Fig. 2j-2l). The result showed that fewer and smaller organoids formed after PRTC treatment. A Western blot result showed that PRTC induced the decrease of CREPT protein in the human organoids. This is consistent with the results of organoids formation from mouse crypt cells under CREPT deletion.

Question: 2. L. 96 and Fig. 1B: Are the crypts isolated from small or large intestine? What does "Intestines" mean: whole pieces of intestine, crypt+villous epithelial cells (e.g. mucosal scrapings), or villous cells only?

Response: Fig. 1b showed qRT-PCR results from small intestines, and the "intestines" means crypts+villous epithelial cells. We isolated the epithelial cells by a dissection protocol. Simply, we put the intestines into EDTA-PBS buffer to allow the epithelial cells disaggregated and then isolate the cells by vigorously shaking. We added the procedure in the revised supplementary materials and methods. We also revised the describing words in the Figure legend of Fig. 1b.

Question: 3. Fig. 1C, D: the smaller size and loss of body weight in the Vil-CREPTKO mice suggest that the digestion and uptake of nutrients in the intestinal villi is strongly reduced. On the other hand, the cell kinetics are changed but the villus number and structure in the jejunum were normal (l. 100 and Suppl. Fig. 1E). To explain this discrepancy, is there evidence for a reduced expression of villus markers (e.g. sucrase-isomaltase; SGLT1; lactase) in the CREPT-KO intestine?

Response: This is a good point. We now further analyzed the RNA-seq data of WT and Vil-CREPT^{KO} intestines. Indeed, the results showed that the expression of genes related to nutrient uptake, such as Slc5a1 (sodium/glucose cotransporter), Slc2a5 (fructose transporter), Sis (sucrase-isomaltase), and Mgam (maltase-glucoamylase) were down-regulated in the

CREPT deleted intestinal epithelial cells. This explained why Vil-CREPT^{KO} mice were leaner but had normal villus number and structure in the jejunum. However, due to the limited space in this single paper, we are sorry we could not be able to present this result. Please refer to the result attached here. We will generate another project to study the down regulation of genes related to nutrient uptake in the CREPT KO mouse in the future. We appreciate this reviewer for this suggestion.

	WT		KO	
	#1	#2	#1	#2
Slc5a1	1640	1547	657	683
Sis	1003	600	459	642
Mgam	668	680	318	347
Slc2a5	245	257	67	73

Figure for reviewer. The FPKM of Slc5a1, Slc2a5, Sis, and Mgam of WT and Vil-CREPT^{KO} intestines from RNA-seq data.

Question: 4. L. 105, Suppl. Fig. 1E: If Ki67 is a reliable marker of proliferating cells, how do the authors explain that the number of Ki67-positive cells in the SI crypts was not different between WT and Vil-CREPTKO mice, whereas CREPT is shown later (l. 153) to be essential for the maintenance of Lgr5+ stem cells and Edu labeling of crypt cells is reduced in Vil-CREPTKO mice (Fig. 1G)?

Response: Ki-67 was widely used as a proliferation marker. However, a previous study showed that more than 10% of Lgr5-GFP+ cells do not express Ki67 using Ki-67 and Lgr5-GFP knock-in mice (Basak et al EMBO J 2012). Their RNA-seq results showed that there was minimal difference between the gene expression in Lgr5^{hi}_Ki67^{low} cells and that in Lgr5^{hi}_Ki67^{hi} cells. The expression level of cell cycle related genes in these two groups of cells were also similar. Their results implied that the Lgr5^{hi}_Ki67^{low} cells are also rapidly proliferating cells. In our case, we observed no difference of the number of Ki67-positive cells in the SI crypts between WT and Vil-CREPT KO mice but indeed a reduced number of Edu labeling of crypt cells in Vil-CREPT KO mice. We speculate that Ki-67 and EdU labelling may represent different states of cell proliferation. EdU labels the fast proliferating cell whose DNA is being synthesized. However, Ki67

staining demonstrates the cells with proliferation ability. In our experiment (Fig. 1f), we found that Edu labeled cells mainly localized at the crypts at 2 h after Edu labeling. However, we observed more Ki-67 positive cells in both the crypt and villi in small intestines (Fig. 1e). Indeed, the Ki-67 positive cells showed no difference in the crypts of the small intestines but decreased in the villi of Vil-CREPT KO mice. Interestingly, we observed that in the large intestine, Ki-67 positive cells were significantly decreased in the crypts in CREPT KO mice (revised supplemental Fig. 1f) but EdU-labeled cells showed no difference (revised supplemental Fig. 1h). All these results suggest that Edu labeling and Ki-67 staining may reflect different stages of cell proliferation. We speculate that CREPT may function on different stages in the small and large intestines. The detailed mechanisms should be deciphered in our future studies.

Question: 5. L. 118, Fig. 1J: the claim that the number of lysozyme-stained Paneth cells is significantly decreased in Vil-CREPTKO mice, if based solely on the “representative” Fig. 1J, is not convincing. Visual inspection suggests that (possibly) the distribution of these cells, but not the number is changed. Is there a drop in the transcript levels of lysozyme or other Paneth cell markers in the intestine of Vil-CREPTKO mice, comparable to the downregulation of Paneth cell markers (Nox 1, Atoh1) reported later (l. 191) in CREPT-deleted Lgr5+ cells?

Response: To address the Paneth cell number alterations, we examined the marker genes. Both the RNA-seq and qRT-PCR results of crypt cells showed significant downregulation of Nox1 and Atoh1 expression in Vil-CREPT^{KO} crypt cells. We now presented the data in revised supplementary Fig. 4f and 4g. We have revised our text to clarify the consistence from the immunostaining and gene expression results for the Paneth cell number changes (see line 228-230).

Question: 6. Fig. 2A: If Bmi1 is robustly expressed not only in +4 reserve stem cells but also in Lgr5+ cells (ref. 16), how do the authors explain that the expression of this ISC marker remains unaltered in Vil-CREPTKO mouse intestine whereas Lgr5+ expression (Fig. 2A) and the number of Lgr5+ expressing ISCs (Fig. 2C) is strongly reduced?

Response: Indeed, *Bmi1* marks not only +4 reserve stem cells but also *Lgr5*⁺ cells. Therefore, it is possible that a part of *Lgr5*⁻ stem cells expresses *Bmi1*. As a similar situation, a study from Metcalfe et al showed that crypts of *Lgr5*⁺ cell deletion were also responding to Wnt signaling under APC deletion (Metcalfe et al, cell stem cell, 2013), suggesting *Lgr5*⁻ stem cells might remain the stem cell features independent of *Lgr5*. We speculate that *Bmi1* might be regulated by both *Lgr5* or non-*Lgr5* pathway. The expression of *Bmi1* may be complemented in *Lgr5*⁻ stem cell when CREPT was deleted. We assume that the loss of *Lgr5*⁺ cells in *Vil-CREPT*^{KO} intestines may cause activation and proliferation of *Lgr5*⁻/*Bmi1*⁺ stem cells. However, we did not have direct evidence to justify this hypothesis at the current study. We would like to address this question in our future projects.

Question: 7. L. 198, Fig. 5A: Expression of the ISC signature genes was barely affected in the (remaining) *Lgr5*⁺-CREPT KO cells. However, as mentioned in l. 246, these stem cell signature genes are mainly regulated by Wnt signaling, which also controls the expression of cell cycle promoting genes and differentiation determining genes. How then can we envision that the latter genes are down-regulated in *Lgr5*⁺-CREPTKO cells, whereas the Wnt-controlled stem cell signature genes are not?

Response: The previous Fig. 5A is now presented as Supplementary Fig. 4a. Indeed, the result showed that ISC signature genes were not affected in the remaining *Lgr5*⁺ CREPT KO cells. We reasoned that this might be due to the remaining cells remains the features of ISC. Interestingly, we observed that the percentage of *Lgr5*⁺ cells were decreased dramatically during the TAM-induced deletion of CREPT (see revised Fig. 2h and 2i). In our *in vitro* organoid culture experiments, we observed that the organoids formed in the first-generation culture from TAM-induced CREPT KO cells failed to survival in the following generation cultures. All these results suggest that the non-synchronized time of CREPT deletion may cause the existence or remaining of *Lgr5*⁺ CREPT KO cells. In another experiment for detecting the gene signatures we directly used the isolated crypts from the wild type and *vil-cre* CREPT KO mice. The results showed that both ISC and Wnt regulated genes were consistently down-regulated under CREPT deletion.

We now discussed the inconsistency of gene signature alterations and updated our presentation in the text (Lines 216-220).

Question: 8. L. 310-321: The surprising finding in this study that Wnt targeting is impaired in regenerating intestine of Vil-CREPTKO mice but does not result in down-regulation of the Wnt target genes CCND1 (Cyclin D1) and c-Myc is interpreted by the authors as a sign of different Wnt signaling regulation between tumors and ISCs (l. 321). It would be of prime importance to know whether this represents a difference between species (mouse vs. human) or between tumors and non-malignant epithelium. In the first case, it would indicate that the mouse model is not an appropriate surrogate for (regenerating) human intestinal epithelium.

In any case, because many other Wnt-target genes are down-regulated in the mouse model, the outcome suggests that CCND1 and MYC do not function as Wnt target genes in mouse intestine. Is there any, more direct evidence supporting this observation?

Response: This is a great point. We have discussed this question in the revised manuscript (line 334-362). We postulate that Wnt-target genes may be differently regulated. In human tumor cells and APC knockout transformed mouse epithelial cells, CREPT upregulates CCND1 and MYC but in normal epithelium, CREPT regulates other Wnt-targeted genes. We performed an additional experiment to show the different regulation of CCND1 and MYC in normal and tumor cells by CREPT. Please refer the result as attached here. Nevertheless, we speculated that Wnt signaling regulation is different between tumors and non-malignant epithelium, not between species (mouse vs. human).

Figure for reviewer. (a and b) Intestinal epithelial cells from *Lgr5-creERT2*; *APC fl/fl* (*Lgr5-APC^{KO}*) and *Lgr5-creERT2*; *APC fl/fl*; *CREPT fl/fl* (*Lgr5-DKO*) mice were used to detect the expression of Wnt target genes, including c-Myc and Cyclin D1. The expression of c-Myc and Cyclin D1 were up-regulated in APC KO cells, but significantly down-regulated in APC-CREPT double knockout cells. (c) CREPT shRNAs were used to deplete the CREPT expression in colorectal tumor cells (DLD1) and non-malignant cells (NCM460) by shRNA, and cyclin D1 expression was detected by a western blotting experiment. The expression of Cyclin D1 was down-regulated by CREPT depletion in DLD1 cells, but that was barely affected in CREPT depleted NCM460 cells.

Question: 9. Fig. 7G: The quantity of nuclear beta-catenin in the crypts of non-irradiated *Vil-CREPT^{KO}* mice was not different from WT mice. On the other hand, the expression of cell cycle promoting genes in *Lgr5-CREPTKO* cells was strongly decreased (Fig. 4G), presumably by a defect in Wnt-beta catenin signaling. Is there any indication that the level of nuclear beta-catenin differs between *Lgr5+*-*CREPTKO* cells and WT *Lgr5+* cells? If not, this would strengthen the notion that the mechanism of Wnt-beta catenin signaling under physiological conditions differs importantly from Wnt-beta catenin signaling in irradiated/regenerating intestine.

Response: This is an excellent point. The previous Fig. 7G is now presented as Supplemental Fig. 6f. We now added another set of staining results in Fig. 6h-6k to present the nuclear β -catenin in the crypts. In the IHC and IF staining images of non-irradiated WT and non-irradiated *Vil-CREPT^{KO}* intestinal section, we could barely observe obvious nuclear b-catenin. To

confirm this observation, we performed an immunofluorescence experiment using the Lgr5⁺ intestinal epithelial cells disaggregated from crypts. The staining result for the β -catenin protein is presented as attached here. It is clear that the nuclear β -catenin was hardly seen in both Lgr5⁺-WT and Lgr5⁺-CREPT^{KO} cells. This supports our notion that Wnt/ β -catenin signaling under physiological conditions differs from the irradiated/regenerating intestine.

Figure for reviewer. Representative images of β -catenin expression in Lgr5⁺ cells. Lgr5-creERT2-GFP (Lgr5-WT) and Lgr5-creERT2-GFP; CREPT^{n/n} (Lgr5-CREPT^{KO}) mice were both intraperitoneally injected with tamoxifen at 1mg a day for 5 consecutive days. Then, Lgr5-WT and Lgr5-CREPT^{KO} cells were harvested by sorting Lgr5^{high} cells at day 7, and attached to slices. Immunofluorescence experiment was performed on the cells to show the β -catenin expression in Lgr5⁺ cells. The nuclear β -catenin was barely observed in both Lgr5-WT and Lgr5-CREPT^{KO} cells. Images are representative of n=3 mice per genotype.

Question: Specific comments (minor):

1. Style and English in the text and figure legends is weak and needs multiple corrections (e.g. CREPT-deficiency mice=CREPT-deficient mice; intestine crypts = intestinal crypts etc.).
2. L. 81: retention in...Please complete the sentence.
3. L. 627: (F and G) should be replaced by(G and H); (H and I) by (I and J).
4. Legends, Suppl. Figs. 4 and 5: Presumably 5 and 2 mM CHIR99021 should be changed to 5 and 2 μ M?

Response: We apologize for these typos and spelling mistakes. We have checked and revised the manuscript.

For 1, we have revised these grammar mistakes.

For 2, the sentence was completed in the manuscript.

For 3, we have checked and revised all the siting figure labels.

For 4, all the units labeling mistakes due to the front conversion were revised.

In addition, we have revised the full text to avoid other similar mistakes.

Thanks to the reviewer again for pointing out these mistakes.

Reviewers' Comments:

Reviewer #1:

Remarks to the Author:

The authors have presented a lengthy and detailed response that addresses many of the points raised not just by myself but by the other reviewers. Where uncertainty remains it is more explicit. The Chip pulldown analysis of CREPT is significant in supporting their interpretation that CREPT regulates Wnt signalling.

Reviewer #3:

Remarks to the Author:

This revised manuscript addresses most of my major concerns, including the issues with the Western Blots. I appreciate the provision of uncropped blots.

I would encourage the author to:

1- Provide a more comprehensive discussion of the differences between the in vivo versus in vitro (organoids) phenotype of Vill-Cre; CREPT KO mice. The authors rightly pointed to compensations by the niche in the in vivo system. Related to that, they should comment on the role of Paneth cells, which are missing upon CREPT knock out.

2- They should mention a limitation of their study (e.g. lack of an experiment with adult specific knock out of CREPT in the intestine. This would have been achieved by the use of inducible Villin- or AhCre-CRE, which I understand they don't have at hand.

3- The new data presented in Figure 6h-K more clearly shows nuclear-B- Catenin. However, the data on Supplementary Figure 6b-e does not and I don't think it adds much to the paper. I suggest removing it.

Reviewer #4:

Remarks to the Author:

The authors have responded adequately to the comments and points of criticism in my report and have modified and replenished their manuscript accordingly. In particular the demonstration that (i) human colorectal organoids seem to respond similarly to CREPT deletion as shown previously for mouse small intestinal organoids (i.e. fewer and smaller organoids formed), strengthening the relevance of the mouse model for human intestine (new Fig. 2j-2l); (ii) the expression of genes related to nutrient uptake were strongly down-regulated in Vil-CREPT-KO mice, explaining why the mice were leaner despite a normal number of villi and structure in the jejunum (the focus of a future project); and (iii) CREPT occupied the promotor and/or termination region of Wnt target genes in mouse intestinal crypt cells (ChIP-seq analysis; new Figs. 5e-f), have contributed importantly to the quality of the model for CREPT function proposed.

There are a few minor points left that need further attention:

1. Details about the origin of the human colonoids and the PRTC treatment protocol are lacking and should be provided in the Method section.
2. L. 221, revised manuscript: "This might be due to remaining Lgr5+ CREPT KO cells remained the feature of ISC to compensate for the loss of Lgr5+ cells." Please clarify this statement.

Reviewer #1 (Remarks to the Author):

The authors have presented a lengthy and detailed response that addresses many of the points raised not just by myself but by the other reviewers. Where uncertainty remains it is more explicit. The Chip pulldown analysis of CREPT is significant in supporting their interpretation that CREPT regulates Wnt signaling.

Responses to Reviewer #1 (Author's Response)

Thank the reviewer for the acceptance.

Reviewer #3 (Remarks to the Author):

This revised manuscript addresses most of my major concerns, including the issues with the Western Blots. I appreciate the provision of uncropped blots. I would encourage the author to:

1- Provide a more comprehensive discussion of the differences between the in vivo versus in vitro (organoids) phenotype of Vill-Cre; CREPT KO mice. The authors rightly pointed to compensations by the niche in the in vivo system. Related to that, they should comment on the role of Paneth cells, which are missing upon CREPT knock out.

2- They should mention a limitation of their study (e.g. lack of an experiment with adult specific knock out of CREPT in the intestine. This would have been achieved by the use of inducible Villin- or AhCre-CRE, which I understand they don't have at hand.

3- The new data presented in Figure 6h-K more clearly shows nuclear-B-Catenin. However, the data on Supplementary Figure 6b-e does not and I don't think it adds much to the paper. I suggest removing it.

Responses to Reviewer #3 (Author's Response)

We thank the reviewer for her/his appreciation of our data and for giving such good suggestions. We have revised our manuscript according to these suggestions:

- 1- We apologize for the lack of consideration of the Paneth cells. Except for mesenchymal cells, Paneth cells secreted several Wnts, including Wnt3, to support the growth of intestinal stem cells (Gregorieff et al., 2005). In purified epithelial cell culture system, Wnt3 from Paneth cells is required for the growth of organoids derived from Lgr5+ stem cells (Ootani et al., 2009; Farin et al., 2012; Sato et al., 2011). However, reduced Paneth cell quantity was observed in CREPT KO intestinal crypts, and the phenotype of organoids was also a consequence of reduced Wnt3 supply. We have added this discussion in our revised manuscript (L.337-347).
- 2- We have discussed this limitation in our revised manuscript. The inducible Villin-Cre or Ah-Cre mice are good for deleting CREPT in adult intestines. Unfortunately, we do not have these mice at hand. We have pointed out this limitation in line 333-335.
- 3- We have deleted the Supplementary Figure 6b-e and updated the related text.

Reviewer #4 (Remarks to the Author):

The authors have responded adequately to the comments and points of criticism in my report and have modified and replenished their manuscript accordingly. In particular the demonstration that (i) human colorectal organoids seem to respond similarly to CREPT deletion as shown previously for mouse small intestinal organoids (i.e. fewer and smaller organoids formed), strengthening the relevance of the mouse model for human intestine (new Fig. 2j-2l); (ii) the expression of genes related to nutrient uptake were strongly down-regulated in Vil-CREPT-KO mice, explaining why the mice were leaner despite a normal number of villi and structure in the jejunum (the focus of a future project); and (iii) CREPT occupied the promoter and/or termination region of Wnt target genes in mouse intestinal crypt cells (ChIP-seq analysis; new Figs. 5e-f), have contributed importantly to the quality of the model for CREPT function proposed.

There are a few minor points left that need further attention:

1. Details about the origin of the human colonoids and the PRTC treatment protocol are lacking and should be provided in the Method section.
2. L. 221, revised manuscript: “This might be due to remaining Lgr5+ CREPT KO cells remained the feature of ISC to compensate for the loss of Lgr5+ cells.” Please clarify this statement.

Responses to Reviewer #4 (Author’s Response)

We are quite pleased the reviewer credited our data convincing. We thank the reviewer for these further suggestions aiming to prove the quality of our manuscript. We have revised our manuscript according to these suggestions:

1. We apologize for missing the important information about human organoids and PRTC treatment. We have added this part in the Methods section of the revised manuscript.
2. We have clarified the statement as “However, as mentioned above, we observed the significant decrease of Lgr5-GFP⁺ cells after TAM induced CREPT deletion (Fig. 2h and 2i). It is possible that the cells with ISC signature genes were survived but other cells were lost in CREPT deleted crypts. The RNA-seq data of Lgr5⁺ cells only reflected the gene expression change in remaining Lgr5+ cells.”